



# Biological and dust aerosol as sources of ice nucleating particles in the Eastern Mediterranean: source apportionment, atmospheric processing and parameterization

Kunfeng Gao[1], Franziska Vogel[2,a], Romanos Foskinis[1,3,4,5], Stergios Vratolis[4], Maria I.Gini[4], Konstantinos Granakis[4], Anne-Claire Billault-Roux[6], Paraskevi Georgakaki[1], Olga Zografou[4], Prodromos Fetfatzis[4], Alexis Berne[6], Alexandros Papagiannis[1,3], Konstantinos Eleftheridadis[4], Ottmar Möhler[2], Athanasios Nenes[1,5]

[1]Laboratory of Atmospheric Processes and Their Impacts, School of Architecture, Civil and Environmental Engineering, École Polytechnique Fédérale de Lausanne, Lausanne, Switzerland
[2]Institute of Meteorology and Climate Research, Karlsruhe Institute of Technology, Karlsruhe, Germany
[3]Laser Remote Sensing Unit (LRSU), Physics Department, National Technical University of Athens, Zografou, Greece
[4]ENvironmental Radioactivity & Aerosol Technology for atmospheric & Climate ImpacT Lab, INRASTES, NCSR Demokritos, 15310 Ag. Paraskevi, Attica, Greece
[5]Centre for Studies of Air Quality and Climate Change, Institute of Chemical Engineering Sciences, Foundation for Research and Technology Hellas, Patras, Greece
[6]Environmental Remote Sensing Laboratory (LTE), School of Architecture, Civil & Environmental Engineering, Ecole Polytechnique Fédérale de Lausanne, Lausanne, Switzerland
[a]Now at: Institute of Atmospheric Sciences and Climate (ISAC), National Research Council (CNR), Bologna, Italy

*Correspondence to*: Athanasios Nenes (athanasios.nenes@epfl.ch) and Kunfeng Gao (kunfeng.gao@epfl.ch)

**Abstract.** Aerosol-cloud interactions in mixed-phase clouds (MPCs) are one of the most uncertain drivers of the hydrological cycle and climate change. A synergy of in-situ, remote sensing and modelling experiments was used to determine the source of ice nucleating particles (INPs) for MPCs at Mount Helmos in the Eastern Mediterranean. The influences of boundary layer turbulence, vertical aerosol distributions and meteorological conditions were also examined. When the observation site is in the Free Troposphere (FT), approximately 1 in $10^6$ aerosol particles serve as INPs. The INP abundance spans three orders of magnitude and increases following the order of marine aerosols, continental aerosols, and finally, dust plumes. Biological particles are important INPs observed in continental and marine aerosols, whereas they play a secondary yet important role even during Saharan dust events. Air masses in the planetary boundary layer (PBL) show both enriched INP concentrations and higher proportion of INPs in comparison to total aerosol particles, different from cases in the FT. The presence of precipitations/clouds enriches INPs in the FT but decreases INPs in the PBL. Additionally, new INP parameterizations, incorporating the ratio of fluorescent-to-nonfluorescent or coarse-to-fine particles and predicting >90% of the observed INPs within an uncertainty range of a factor of 10, exhibit better performance than current widely-used parameterizations, and allow ice formation in models to respond to variations in dust and biological particles. The improved parameterizations can help MPC formation simulations in regions with various INP sources or different regions with prevailing INP sources.





## 1 Introduction

Clouds in the atmosphere can be either composed solely of liquid water droplets, ice crystals, or a mixture of the both (mixed-phase clouds; MPCs). Cloud phase regulates the optical properties and microphysical characteristics of the clouds, which further influences their impacts on the hydrological cycle and the climate (Tan et al., 2016; Lohmann and Neubauer, 2018; Zhou et al., 2022). Notably, the modulation of cloud properties from anthropogenic particles is one of the leading sources of uncertainty in anthropogenic climate change (e.g., Seinfeld et al. (2016)). MPCs are ubiquitous (D'alessandro et al., 2019) but have a much more uncertain impact on the climate compared to single-phase clouds (Bjordal et al., 2020). This uncertainty stems from the large number of interactions that can take place between liquid droplets, ice crystals, and water vapor - each of which can cool or warm the climate. Furthermore, MPCs exhibit considerable dynamic variability over time and location because they are thermodynamically unstable. Under supercooled conditions, the saturation vapor pressure with respect to ice ($S_i$) is higher than that with respect to water ($S_w$), which favours the mass transfer – through deposition from the vapor phase – of liquid water from particles onto ice existing in MPCs, i.e., the growth of the latter at the expense of the former. This process is known as the Wegener-Bergeron-Findeisen (WBF) process (Wegener, 1912; Bergeron, 1935; Findeisen, 1938; Findeisen et al., 2015). The amount of ice crystals in MPCs is also regulated by the abundance of aerosol particles capable of initiating ice formation, i.e., ice nucleating particles (INPs). INPs can trigger primary ice formation in the MPC regime (Kanji et al., 2017; Burrows et al., 2022; Knopf and Alpert, 2023) with the absence of spontaneous ice formation via the homogeneous freezing of solution droplets which otherwise requires temperatures ($T$) lower than the homogeneous nucleation temperature (Barahona and Nenes, 2009; Lohmann et al., 2016). Thus, the heterogeneous ice nucleation of INPs can lead to MPC glaciation. An added complexity is ice multiplication processes (or secondary ice processes, SIP) occurring in warmer MPCs, which can multiply ice crystal numbers by orders of magnitude above the primary ice levels generated from INPs (Field et al., 2017; Sullivan et al., 2018; Georgakaki et al., 2022; Pasquier et al., 2022). Constraining therefore the abundance and origin of INPs is critical for understanding MPCs formation and their effects on the hydrological cycle and climate.

MPCs persistently exist in mountainous terrain where local and remote air masses may be present (Pousse-Nottelmann et al., 2015; Lohmann et al., 2016; Henneberg et al., 2017). Different air masses, e.g., continental pollution, dust plumes, and sea spray aerosols from remote marine areas, may contain distinct INP populations with characteristic abundance and ice formation ability (Demott et al., 2010; Tobo et al., 2013; Mccluskey et al., 2018; Brunner et al., 2021). The INP type, which depends on its air mass source, is crucial for determining MPC formations. Different types of INPs nucleate ice in different $T$ regimes. Biological particles as effective INPs are active for $T$ warmer than −15°C whereas dust particles generally form ice for $T$ lower than −15°C (Murray et al., 2012). In addition, the formation and evolution of MPCs over orographic terrains are influenced by the planetary boundary layer height (PBLH) (Miltenberger et al., 2020), which regulates the aerosol sources depending on whether the observation site is inside or outside the planetary boundary layer (PBL) (Conen et al., 2015; Wieder et al., 2022). In-situ observations at high altitudes in mountainous terrains provide the possibility to specifically investigate INP populations





relevant for MPCs under different atmospheric conditions, given that the relative position of mountain top in the atmosphere can vary with changing PBLH (Foskinis et al., 2024, under review).

We study the source of INPs and the characteristics of different INP sources relevant for orographic MPCs in the Eastern
Mediterranean region. A field campaign, the Cloud-AerosoL InteractionS in the Helmos background TropOsphere (CALISHTO), was conducted at the Helmos Hellenic Atmospheric Aerosol and Climate Change (in short as (HAC)[2] hereafter) station (37.9843° N, 22.1963° E, 2314 m above sea level (a.s.l.)) close to the summit of Mt. Helmos. It is reported that (HAC)[2] is a station among 12 in Europe with the lowest impacts from the PBL (Collaud Coen et al., 2018), suggesting that (HAC)[2] is an appropriate station to study INPs from remote sources and to evaluate the characteristics of INPs under background
conditions, e.g., in the free troposphere (FT).

This study presents the observations of INPs and aerosol properties at (HAC)[2] during CALISHTO campaign from a period between October 12 and November 27 in 2021. The objectives of this study are twofold: (i) to identify different INP sources at Mt. Helmos and evaluate the characteristics of these INP sources. To this end, a synergy of in-situ aerosol property measurements, remote sensing measurements and model simulations of air mass trajectories, was used to identify different
INP sources. In addition, the influence of precipitation and/or clouds on the INP characteristics was also investigated, considering that precipitations or clouds may serve as sinks or sources of aerosol particles (Isokääntä et al., 2022; Khadir et al., 2023), which therefore influence the INP abundance of the source. (ii) to use the data and analysis carried out to evaluate existing INP parameterizations and propose new ones that capture the INP number concentration ($N_{INP}$) better for the wide diversity of particle types encountered at Mt. Helmos during CALISHTO. Compared to different parameterizations reported
in the literature (Demott et al., 2010; Niemand et al., 2012; Tobo et al., 2013; Demott et al., 2015; Ullrich et al., 2017; Mccluskey et al., 2018), the improvement of new INP parameterizations is based on the advantage of the inclusion of source characteristics, i.e., the partitioning of fluorescent and non-fluorescent (or coarse and fine) particles, yielding important implications for modelling studies aimed at quantifying climate effects of cloud-aerosol interactions in MPCs.

## 2 Methods

### 2.1 Overview of field campaign

(HAC)[2] is an atmospheric monitoring station, contributing data to GAW and ACTRIS since 2016 (Laj et al., 2020), located near the summit of Mt. Helmos at the heart of the Peloponnese in Greece. (HAC)[2] is frequently situated in the FT or at the FT/PBL interface (Foskinis et al., 2024, under review) while it is also frequently covered by clouds in the Fall and springtime, which allows for the in-situ study of aerosol-cloud interactions for warm and MPC clouds. (HAC)[2] is also located at a cross-
road of different air masses, including continental pollution, Saharan dust events, long-range transported biomass burning, marine sea spray aerosols, etc, which allows one to explore the effects of different aerosol types on cloud formation, and which are explored here. The experimental setup of the CALISHTO observations is presented in Fig. 1, including in-situ ice nucleation (IN) experiments, aerosol properties (size distribution, chemical composition, fluorescent and optical properties) at





(HAC)², remote sensing measurements conducted at Vathia Lakka (VL) a site 500 m lower than (HAC)², and back trajectory
analysis for calculating the origin of air masses sampled at (HAC)². Also, meteorological standard parameters recorded at
(HAC)² were used to correct the measured INP and aerosol particle number concentrations to values under the equivalent
atmospheric standard condition (i.e., per standard volume of sampled air, std).

**Figure 1. Overview of the instrumentation setup for CALISHTO campaign. 1: (HAC)² stands for the mountaintop station where in-**
**situ measurements were performed, including ice nucleation measurements and aerosol property measurements: Portable Ice**
**Nucleation Experiment (PINE), Scanning Mobility Particle Sizer (SMPS), Aerodynamic Particle Sizer (APS), Wideband Integrated**
**Bioaerosol Sensor (WIBS), nephelometer and aethalometer, as well as Time-of-Flight Aerosol Chemical Speciation Monitor (ToF-**
**ACSM). Filters collected for off-line analysis by using Ice Nucleation Spectrometer of the Karlsruhe Institute of Technology**
**(INSEKT). 2: VL represents the Vathia Lakka site at the base of Mt. Helmos, at an altitude of ~1.8 km, at which a HALO wind lidar**
**and a frequency-modulated continuous wave (FMCW) Doppler radar (working at 94 GHz) were placed for remote sensing of wind**
**fields, aerosols and clouds. 3: Modelling products include FLEXible PARTicle dispersion model (FLEXPART) to determine the**
**source regions of aerosol particles reaching the site, Hybrid Single-Particle Lagrangian Integrated Trajectory (HYSPLIT) to acquire**
**airmass atmospheric trajectories, and the SKIRON model to obtain dust forecasts. PBL is the planetary boundary layer and FT is**
**the free troposphere. PBLH stands for PBL height relative to the VL.**



## 2.2 INP observations

### 2.2.1 Offline INP observation

The Ice Nucleation Spectrometer of the Karlsruhe Institute of Technology (INSEKT) freezing assay (Schiebel, 2017; Schneider et al., 2021) was used to measure the immersion mode INPs from 0 to −25 °C (Fig. 1). INSEKT measures the freezing $T$ of small water volumes (50 µL) that contain aerosol suspended in them. To prepare the freezing aliquots, aerosol particles were first sampled onto filters from an omnidirectional total inlet at $(HAC)^2$ and then extracted in nano-pure water which was beforehand filtered through a 0.1 µm Whatman syringe filter. The aerosol suspension was diluted by 15:1 and 225:1using two different ratios before pipetting into two 96-well polymerase chain reaction (PCR) plates. In addition to the original solution and the dilutions, some of the wells (~32) are filled with nano pure water for freezing background tests. The PCR plates were then placed in an aluminium block cooled by an ethanol chiller to perform freezing experiments and the frozen fraction of the prepared aliquots was recorded as a function of $T$. Following the analysis protocol reported in Vali (1971) and Vali (2019), the INP concentration of aerosol samples (in $L^{-1}$) as a function of $T$ can be calculated using the tested frozen fraction, sampled aerosol volume, suspended liquid volume and the dilution ratio. The sampling time of each filter sample was approximately 24 hours and a few filter samples had a longer sampling time; detailed information for each INSEKT filter will be provided in the overview paper for CALISHTO campaign.

### 2.2.2 Online INP observations

A portable ice nucleation experiment (PINE) (Möhler et al., 2021) chamber was used for automated real-time observations of INPs at $(HAC)^2$ (Fig. 1). PINE is designed on the basis of expansion cooling of air parcels (Mohler et al., 2003), where ambient air (10 L) is sampled into a pre-cooled cloud chamber after passing through Nafion dryers to remove excess moisture and avoid chamber frosting. The air is then expanded and cools down until supersaturation is reached, causing the aerosol particles in the sample to form supercooled droplets and/or ice crystals. Both the number concentration and phase of aerosol particles are monitored, so INPs that activate to ice crystals can be differentiated from droplets and counted. PINE can be operated in repeated cycles by refilling the cloud chamber with fresh aerosol at the end of an expansion. In this study, PINE was operated in a $T$ range from −23 to −28 °C and $S_w$ (saturation with respect to water) >1.0 to measure INPs activating as ice in the immersion freezing mode. A single PINE expansion cycle has a time resolution of 6 min and the INP number concentration detection limit for a single experiment is approximately 0.5 $L^{-1}$; averaging over an hour of samples reduces the detection limit to approximately 0.05 $L^{-1}$ (Möhler et al., 2021). The same sampling inlet was used for both PINE and INSEKT.

## 2.3 Aerosol property measurements

### 2.3.1 In-situ aerosol property monitoring

Ambient air was sampled through a total inlet for in-situ aerosol property measurements at $(HAC)^2$ (Fig. 1). A downstream inlet with an impaction stage supplies PM10 aerosol to a scanning mobility particle sizer (SMPS; Vienna-type differential



mobility analyser (DMA), and condensation particle counter (CPC) 3772, TSI Inc., US) and an aerodynamic particle sizer (APS; model 3321, TSI Inc., US) to measure the aerosol particle size distribution range from 10 to 800 nm (electrical mobility diameter) and 0.5 to 20 µm (aerodynamic diameter), respectively. The number concentration of aerosol particles larger than 95 nm (SMPS $N_{>95nm}$) was calculated from the SMPS data and used with a threshold value (100 std cm$^{-3}$) to determine if

(HAC)$^2$ is inside or outside the PBL (Herrmann et al., 2015; Brunner et al., 2021), given that large-sized aerosol particles are much rarer outside the PBL. This threshold is confirmed by other methods determining the PBLH using the wind lidar and other data (Foskinis et al., 2024, under review). The number concentrations of total, coarse (>1.0 µm, aerodynamic diameter) and fine (<1.0 µm) particles were also calculated based on APS data, termed Total$_{APS}$, Coarse$_{APS}$ and Fine$_{APS}$, respectively. In addition, the combined aerosol particle distribution observed by both SMPS and APS was calculated by using the method

reported in Khlystov et al. (2004). Accordingly, the number concentrations of total particles (Total$_{SMPS+APS}$) and total fine particles (Fine$_{SMPS+APS}$ <1.0 µm, aerodynamic diameter) observed by both SMPS and APS were also calculated.

A wideband integrated bioaerosol sensor-New Electronics Option (WIBS-5/NEO, Droplet Measurement Technologies, LLC. US) was used at (HAC)$^2$, sampling aerosols with a PM10 impactor, to measure the size distribution of aerosol particles with optical sizes larger than 0.5 µm. WIBS also measures the fluorescence of aerosols on a single particle basis using ultraviolet

light to trigger the excitation of the particle and then detecting the fluorescence at three fluorescent channels: FL1 (excitation wavelength at 280 nm and emission detection at the waveband 310–400 nm), FL2 and FL3 (emission detection waveband of 420–650 nm probing particles excited at 280 and 370 nm respectively). These three channels target at different biologic fluorophores: tryptophan-containing proteins, NAD(P)H co-enzymes and riboflavin, respectively (Kaye et al., 2005; Savage et al., 2017), which are ubiquitous in microbes (Pöhlker et al., 2012). Particles showing fluorescence exclusively in any one of

three channels are attributed to a type of A$_{WIBS}$ (FL1 only), B$_{WIBS}$ (FL2 only) or C$_{WIBS}$ (FL3 only), respectively. Particles carrying two types of fluorophores and simultaneously detected by two channels are termed AB$_{WIBS}$ (FL1 and FL2), AC$_{WIBS}$ (FL1 and FL3) or BC$_{WIBS}$ (FL2 and FL3). Particles showing fluorescence in any one of the three channels are termed Fluo$_{WIBS}$. Particles detected in all three channels are attributed to a type of ABC$_{WIBS}$, which are much more likely to be of biological origin compared to the other types (Hernandez et al., 2016; Savage et al., 2017). Note that non-biological particles may also

present in B$_{WIBS}$, C$_{WIBS}$ and BC$_{WIBS}$ channels, behaving as interfering particles such as some black carbon and dust particles associated with fluorescent materials (Toprak and Schnaiter, 2013; Savage et al., 2017). The fluorescence detection limit was determined by subtracting the mean background signal plus 9× standard deviations measured from routinely forced trigger tests. The WIBS data can be resampled to a customized time span because of its 15 µs high time resolution for single particle detection. The measurement rates of WIBS-5/NEO are up to 9500 cm$^{-3}$ for all particles irrespective of fluorescence and up to

466 cm$^{-3}$ for fluorescent particles.

The hourly mean light-scattering coefficient of dry PM10 aerosols at (HAC)$^2$ was measured by using an integrating nephelometer (Model 3563, TSI Inc., US) at 3 wavelengths (450, 550 and 700 nm, termed Scatt$_{450\,nm}$, Scatt$_{550\,nm}$ and Scatt$_{700\,nm}$) (Laj et al., 2020). Using light scattering coefficients measured at 450 and 700 nm wavelength, the Ångström exponent (α) can be calculated following Eq. (1):




$$\alpha = -\frac{\ln\left[Scatt_{700\,nm}/Scatt_{450\,nm}\right]}{\ln(700/450)}. \quad (1)$$

The wavelength pair of 450 and 700 nm was used because the larger difference in the measured scattering coefficients gives more accurate α values (Mordas et al., 2015). A lower α value suggests the dominance of coarse particles in the sampled aerosols whereas a larger α value indicates the dominance of fine particles (Pereira et al., 2008), which helps to differentiate continental aerosols mainly containing fine particles from dust plumes dominated by coarse particles. In addition, the mass

concentration of refractory and carbonaceous aerosol particles, i.e., elemental black carbon (eBC), sampled through a PM10 cut-off inlet was monitored by an aethalometer (AE31, Magee Scientific, US) at 880 nm with a minimum time base of 2 min. The chemical composition of non-refractory species of submicron ambient aerosols, including organics (Org), sulphate ($SO_4^{2+}$), nitrate ($NO_3^-$), ammonium ($NH_4^+$) and chloride ($Cl^-$), was monitored by a time-of-flight aerosol chemical speciation monitor (ToF-ACSM, Aerodyne Research Inc., USA) with a time resolution of 10 min (Zografou et al., 2024, in preparation).

**2.3.2 Airmass remote sensing measurements**

Figure 1 shows that remote sensing measurements were performed at VL at an altitude lower than (HAC)[2] by 500 m to measure the PBLH, radar equivalent reflectivity factor (Ze) and mean Doppler velocity (MDV). The PBLH was calculated by using the Doppler velocity of aerosols measured by a pulsed Doppler scanning lidar system (StreamLine Wind Pro model, HALO Photonics, UK). The lidar was operated in the stare vertical azimuth display mode at a wavelength of 1.5 μm with time

resolution of 10 min and a vertical spatial resolution of 30 m. The vertical wind speed distribution at a certain distance from the lidar was calculated (Barlow et al., 2011; Schween et al., 2014). The PBL top boundary is defined at a position where the standard deviation of the wind vertical velocity, $\sigma_w$, drops below 0.1 m$^2$ s$^{-2}$ (Foskinis et al., 2024, under review). The vertical distance between VL and the PBL top boundary then is the PBLH. Note that PBLH may be unavailable during cloudy periods because the lidar signal is quickly attenuated in clouds and when too few particles are present because insufficient

backscattering signal will inhibit PBLH determination. When PBLH results are unavailable, SMPS $N_{>95nm}$ results were compared to the threshold value (100 std cm$^{-3}$) to define the (HAC)[2] position with respect to PBL (Herrmann et al., 2015; Brunner et al., 2021). In addition, a frequency-modulated continuous wave (FMCW) Wide-band Doppler spectral zenith profiler (WProf) was deployed to measure the radar reflectivity at a wavelength of 3.2 mm (corresponding to 94 GHz) up to 10 km above the ground level (a.g.l., Küchler et al., 2017; Billault-Roux et al., 2023; Ferrone and Berne, 2023). The Doppler

radar results at (HAC)[2] level (500 m above VL) were used to evaluate the presence of precipitation and clouds.

**2.3.3 Airmass modelling**

As shown in Fig. 1, the footprint and trajectory of airmasses arriving at (HAC)[2] were calculated by the FLEXPART (FLEXible PARTicle dispersion Model) (Stohl et al., 2005; Pisso et al., 2019; Vratolis et al., 2023) and the HYSPLIT (Hybrid Single-Particle Lagrangian Integrated Trajectory) (Draxler and Hess, 1998; Stein et al., 2015) models. FLEXPART was used to

calculate the residence time of aerosol particles with a geometric mean diameter of 400 nm (10 nm − 10 μm, Fig. S1) and a



standard deviation of 3.3 in defined locations in the past 10 days. The spatial resolution of the model corresponds to a grid cell size of 1°×1°. Note that more than 90% of the aerosol particles used in FLEXPART simulations have diameters larger than 100 nm and particles of this size range (> 100 nm) are mainly responsible for the observed INPs. The 24-hour aerosol footprint simulation for each calendar day was run every 3 hours (from 00:00 to 24:00 LTC) by releasing 40000 air parcels from (HAC)[2].

Local wind and turbulence, as well as mesoscale wind fluctuations were considered in the dispersion and transport calculations. Dry and wet aerosol deposition processes were also included in the model. The residence time of aerosol particles in each location was integrated from a height of 0 to 500 m a.g.l.

The HYSPLIT was run to calculate the 7-day backward trajectories of air masses arriving at (HAC)[2]. Input meteorological data from the Global Data Assimilation System (GDAS, 1°×1° resolution) (Stein et al., 2015; Kostrykin et al., 2021) were

used for the backward trajectory calculations. The source height was set at three height levels of 100, 1000 and 2000 m a.g.l. The model was launched every 6 h backward the start time. The start time of back trajectories on a day was decided depending on the need to differentiate whether there is an aerosol source change during the day.

Finally, the SKIRON model (Kallos et al., 2006; Spyrou et al., 2010) was used to calculate the time series of dust mass concentration at (HAC)[2] at different height levels of 1250, 1614, 1881 and 2170 m a.s.l. Together with Coarse$_{APS}$ particle

concentration and the aerosol footprints from FLEXPART, dust mass concentrations below (HAC)[2] predicted by SKIRON help to determine the occurrence and intensity of dust plumes and their source region.

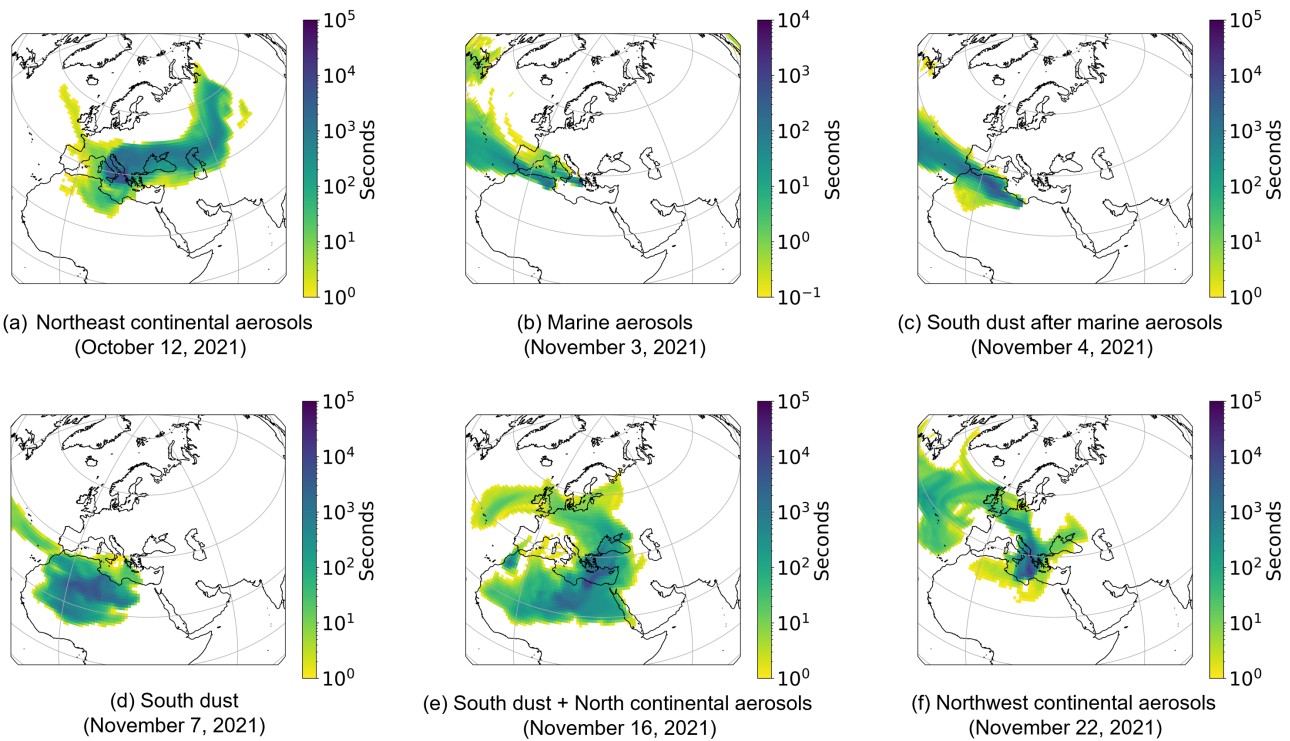

**Figure 2. Overview of aerosol sources as seen from exemplary FLEXPART 10 day backward residence time maps. The colour map scales the residence time of the particles from a height between 0 and 500 m a.g.l. in the 10 d back trajectories. (a) Northeast**



**continental air masses on October 12, 2021. (b) Marine air masses on November 3, 2021. (c) South dust after marine air masses on November 4, 2021. (d) South dust on November 7, 2021. (e) South dust + North continental air masses on November 16, 2021. (f) Northwest continental air masses on November 22, 2021.**

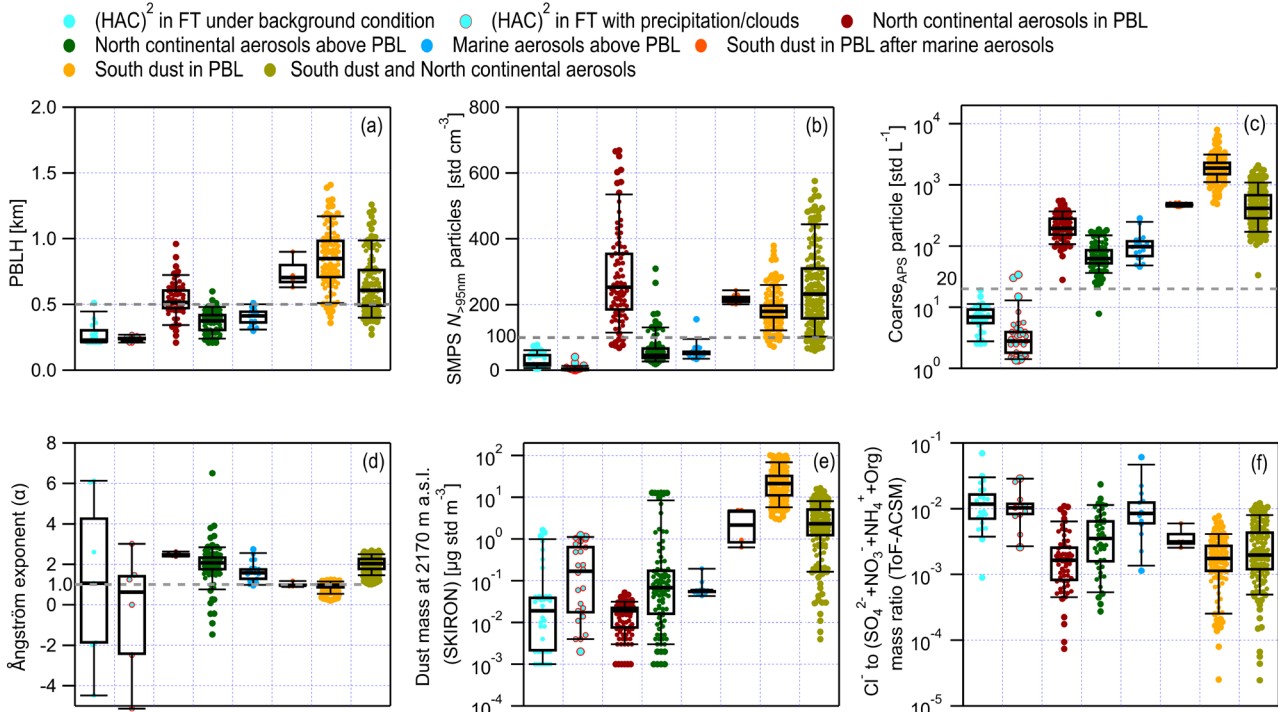

**Figure 3. Box plots for the characteristics of identified aerosol sources. (a) PBLH with respect to VL. (b) Number concentration of**
**particles larger than 95 nm measured by SMPS (SMPS $N_{>95nm}$) at (HAC)$^2$. (c) Coarse particle (>1.0 μm) number concentration measured by APS (Coarse$_{APS}$) at (HAC)$^2$. (d) Ångström exponent (α) at the wavelength pair of 450 and 700 nm calculated using nephelometer data recorded at (HAC)$^2$. (e) Dust mass concentration at 2170 m a.s.l. (~140 m below (HAC)$^2$) calculated by the SKIRON model. (f) The mass ratio of Cl$^-$ to other species measured by ToF-ACSM at (HAC)$^2$. The box shows the median line and the range between 25$^{th}$ and 75$^{th}$ quartiles. The lower and upper caps of the box indicate the 9$^{th}$ and 91$^{th}$ percentiles, respectively.**

**2.4 Aerosol source apportionment and type classification**

In general, transported aerosol particles from remote regions showed trajectories from the north in earlier October at the beginning of the campaign and changed counter clockwise during the campaign. Figure 2 presents exemplary FELXPART results throughout the campaign to show major aerosol sources from remote regions. Aerosol particles arriving at (HAC)$^2$ on October 12 (Fig. 2a) came from the north and the northeast, which can be attributed to continental aerosols, while later on in

November 3, marine aerosols from the Atlantic Ocean and the Mediterranean Sea might take a large fraction of aerosols transported to (HAC)$^2$ (Fig. 2b). On November 4 and later, dust from the Sahara, possibly mixed with marine aerosols reach the station (Fig. 2c and 2d). Notably, the Saharan dust event increases the aerosol content (Fig. 2c) by more than one order of magnitude compared to the marine aerosols (Fig. 2b). At the beginning of the Saharan dust event, the case in Fig. 2c spanned ~3 hours which might be a transition period with a lower mixture of local aerosols. After that, Saharan dust (Fig. 2d) persisted




for about one week. Later on, dust mixed with continental particles reached the site (Fig. 2e), followed by primarily continental aerosols from the North of (HAC)$^2$, e.g., Balkans (Fig. 2f). Note that FLEXPART results provide an overview of aerosol sources on a daily basis, the further identification of particle sources at (HAC)$^2$ relies on both in-situ and remote sensing results with a time resolution of one hour. For example, the synergy of in-situ and remote sensing results enables to differentiate the difference between the sources in Fig. 2b and c.

In addition to aerosol footprints, we consider following criteria to specifically classify the sources of air masses at (HAC)$^2$:

- Comparing PBLH (SMPS $N_{>95nm}$) with a threshold value of 0.5 km (100 std cm$^{-3}$) to examine the relative position of (HAC)$^2$ with respect to the PBL

- Comparing Coarse$_{APS}$ with a threshold value of 20 std L$^{-1}$ to judge the presence of remotely transported air masses in the FT

- Comparing Ze (MDV) with a threshold value of 10 dBZ (−0.5 m s$^{-1}$) to evaluate the presence of precipitation/clouds

- Comparing α with a threshold value of 1.0 to diagnose the occurrence of Saharan dust events

Figure 3 summarizes the classified aerosol sources and presents their characteristics to demonstrate their distinct nature as follows. Hourly-averaged data is presented in Fig. 3, however, presented data for the source of South dust in PBL after marine aerosols in Fig. 3 and following figures is resampled for every 15 minutes due to a short period of observation (< 3 hours).

First, a PBLH less than 0.5 km or SMPS $N_{>95nm}$ less than 100 std cm$^{-3}$ (if no PBLH results available) means (HAC)$^2$ is above the PBL (Fig. 3a or b). Moreover, if Coarse$_{APS}$ is less than 20 std L$^{-1}$, the period will be attributed to (HAC)$^2$ in FT (Fig. 3c). Furthermore, periods of Ze values larger than 10 dBZ (Hagen and Yuter, 2006) and MDV less than −0.5 m s$^{-1}$ are classified as periods influenced by precipitation. For periods with slightly lower Ze values, (HAC)$^2$ is likely in-cloud or fogs. During the campaign, periods of precipitation frequently alternated with cloudy periods. Therefore, we consider such periods jointly. If

there is no influence from remotely transported air masses (Coarse$_{APS}$<20 std L$^{-1}$), depending on the presence of precipitation/clouds, the condition is classified as (HAC)$^2$ in FT under background condition and (HAC)$^2$ in FT with precipitation/clouds, respectively. For periods of Coarse$_{APS}$>20 std L$^{-1}$, the influence of remotely transported aerosols was considered. Continental aerosols have α values larger than 2.0 (Fig. 3d) when (HAC)$^2$ is in the PBL. The distinct particle properties of continental aerosols also include high SMPS $N_{>95nm}$ (median value >200 std cm$^{-3}$ in Fig. 3b), moderate Coarse$_{APS}$

values (median value ~200 std L$^{-1}$ in Fig. 3c) and very low dust particle abundance (<0.1 µg std m$^{-3}$ in Fig. 3e). Such an aerosol source is termed North continental aerosols in PBL. When continental aerosols are sources of particles at (HAC)$^2$ but the site is above the PBL (Fig. 3a and 3b), the 75$^{th}$ quartile for Coarse$_{APS}$ of the aerosols decreases to 91 std L$^{-1}$ (Fig. 3c) which becomes lower than the 9$^{th}$ percentile for Coarse$_{APS}$ of North continental aerosols in PBL. Also, the α value of the source termed North continental aerosols above PBL decreases (25$^{th}$ quartile >1.67) but the median value is still higher than 2.0 (Fig.

3d). Therefore, North continental aerosols above PBL are distinctly different from the aerosols in the PBL. In addition, the distinction of marine aerosols above PBL is indicated by the highest Cl$^-$ fraction compared to the other scenarios (Fig. 3f). Khan et al. (2015) reported that the particle number concentration of coarse mode sea spray aerosols is less than that of dust aerosols by approximately 100 times, which is consistent with the observations in this study (Fig. 3c). Hence, a scenario of

 

Marine aerosols above PBL can be classified. Followed by marine aerosols, a distinct period of several hours (on November

4) at the beginning of a dust event (from November 4 to 10) was observed for (HAC)² in the PBL. It shows high Coarse_APS particle concentrations (>450 std L⁻¹ in Fig. 3c), low α values (close to 1.0 in Fig. 3d) and the presence of dust particles (Fig. 3e). Thus, such a source is classified as South dust in PBL after marine aerosols. Afterwards, the dust event period is termed South dust in PBL which shows the highest Coarse_APS particle concentration (9ᵗʰ percentile >1000 std L⁻¹ in Fig. 3c), low α values (close to 1.0, Fig. 3d) and the largest dust mass concentration (25ᵗʰ quartile >10 μg std m⁻³ in Fig. 3e). Lastly, periods

of aerosol footprints similar to Fig. 2e and showing aerosol properties in Fig. 3 between those of South dust and North continental aerosols are classified as South dust with North continental aerosols, i.e., a mixture of both aerosols. Hereafter, we use the remotely transported aerosols identified in Fig. 3 to name the aerosol sources at (HAC)². However, we note that particles from local sources may also be relevant for aerosol particles reaching (HAC)² depending on the (HAC)² position with respect to the PBL.

**Table 1. INP parameterizations from the literature.**

| INP Parameterization from literature | Region | Included major aerosol types | T range of INP observations | Formulation |
|---|---|---|---|---|
| DeMott2010 | Global observation covering Colorado, Wyoming and Alaska in US, Eastern Canada and Ottawa, Pacific region and Amazon basin | Dust and biological particles | −35 – −9 °C ($S_w$ > 100%) | $N_{INP} = a(-T)^b * Total_{APS}^{(-cT+d)}$ (a = 0.0000594, b = 3.33, c = 0.0264, d = 0.0033) ($Total_{APS}$ in std cm⁻³; T in °C) |
| Tobo2013FBAP | Rocky Mountain region | Biological particles | −35 – −9 °C ($S_w$ = 103–106%) | $N_{INP} = Fluo_{WIBS}^{(-aT+b)} * exp(-cT+d)$ (a = −0.108, b = 3.8, c = 0, d = 4.605) ($Fluo_{WIBS}^@$ in std cm⁻³; T in °C) |
| DeMott2015 | Pacific Ocean basin and Virgin Islands | Laboratory dust samples and dust particles in the atmosphere | −35 – −20 °C ($S_w$ = 105%) | $N_{INP} = cf * Total_{APS}^{(-aT+b)} * exp(-cT+d)$ (cf = 3, a = 0, b = 1.25, c = 0.46, d = −11.6) ($Total_{APS}$ in std cm⁻³; T in °C) |
| Niemand2012 | Laboratory experiments | Dust | −36 – −12 °C ($S_w$ > 100%) | $N_{INP} = 1000 * S_{SMPS+APS} * exp(aT+b)$ (a = −0.517, b = 8.934) ($S_{SMPS+APS}$ in std m² cm⁻³; T in °C) |
| Ullrich2017 | Laboratory experiments | Dust | −30 – −14 °C ($S_w$ > 100%) | $N_{INP} = 1000 * S_{SMPS+APS} * exp(a(273.15+T)+b)$ (a = −0.517, b = 150.577) ($S_{SMPS+APS}^#$ in std m² cm⁻³; T in °C) |
| McCluskey2018 | West coast of Ireland | Sea spray aerosols, marine organics and offshore biological particles | −27 – −10 °C ($S_w$ > 100%) | $N_{INP} = 1000 * S_{SMPS+APS} * exp(aT+b)$ (a = −0.545, b = 1.0125) ($S_{SMPS+APS}$ in std m² cm⁻³; T in °C) |



[@] Fluo$_{WIBS}$ means the number concentration of fluorescent particles larger than 0.5 μm measured by WIBS. Note that Tobo et al. (2013) used UV-APS to measure fluorescent particles showing fluorescence signals in the wavelength range of 400−575 nm after being excited at 355 nm

[#] S$_{SMPS+APS}$ total particle surface area measured by SMPS and APS

## 2.5 INP parameterization methods

INP parameterization is critically important for climate models to express cloud-aerosol interactions in both ice clouds and MPCs. As a minor subset of total aerosol particles, $N_{INP}$ exponentially increases with decreasing $T$, as supported by the theory (Kampe and Weickmann, 1951) and observations (reviewed in Kanji et al., 2017). $N_{INP}$ also shows dependence on the concentration of available aerosol particles (Burrows et al., 2022) and characteristic of their surfaces (e.g., IN active site density), to trigger the activation (Vali et al., 2015; Knopf and Alpert, 2023). Demott et al. (2010) developed a parameterization (termed DeMott2010 in Table 1) to predict $N_{INP}$ using $T$ and the number concentration of aerosol particles larger than 0.5 μm (Total$_{APS}$), based on a suite of INP measurements at various locations globally. Tobo et al. (2013) augmented the DeMott2010 formulation and developed a new parameterization (termed Tobo2013FBAP in Table 1) to calculate $N_{INP}$ using the number concentration of fluorescent aerosol particles monitored by an UV-APS (Ultra-violet APS, $n_{FBAPs}$), to consider the explicit contributions from biological and dust particles in the coarse mode population. Compared to DeMott2010, Tobo2013FBAP shows increased predictability to calculate $N_{INP}$ from aerosol sources containing biological particles (Tobo et al., 2013). Demott et al. (2015) used Total$_{APS}$ and augmented Tobo2013FBAP by introducing a calibration factor ($cf$) and calculated a new suite of parameters for the formulation by fitting to integrated laboratory and field data (termed DeMott2015 in Table 1). However, the $N_{INP}$ of different aerosol sources may not scale to the total aerosol particle number concentration (e.g., Total$_{APS}$ or $n_{FBAPs}$) following the same rule as used in the above-mentioned parameterizations. This is because the IN ability of potential INPs from different sources varies and different types of INPs dominate the $N_{INP}$ in different $T$ regimes (Murray et al., 2012; Kanji et al., 2017).

Particle-surface-area based approaches were also reported in the literature, such as approaches termed Niemand2012 (Niemand et al., 2012), Ullrich2017 (Ullrich et al., 2017) and McCluskey2018 (Mccluskey et al., 2018) in Table 1. Given that different types of INPs originate from different sources may have different IN active site densities over the particle surface, the INP concentrations calculated from different particle-surface-area based approaches developed from different aerosol sources can vary by more than three orders of magnitude (Niemand et al., 2012; Mccluskey et al., 2018). Further improvements in INP prediction may require parameterizations to explicitly consider additional characteristics of resolved aerosol properties to point to their sources. Mignani et al. (2021) reported that the ratio of large-sized aerosol particles (>2.0 μm) recorded by APS can be used as an identity to characterize INPs from Saharan dust and the implementation of the ratio into the INP parameterization improves its predictability. We expand upon this and use the detailed INP source apportionment and type classification (Section 2.4) to incorporate INP source characteristics into INP parameterizations developed here. The new parameterizations proposed for INPs at Mt. Helmos are expressed as a function of $T$, aerosol particle concentration recorded by APS or WIBS, and the





ratio of particle number concentration recorded in different channels of APS or WIBS. We demonstrate the feasibility of
predicting INPs from different sources using one suite of parameters for the proposed parameterizations (see Section 3.4), as
well as the superior performance of the new parameterizations compared to the approaches reported in the literature (Table 1).

## 3 Results and Discussions

In this section, we first provide evidence on the distinct characteristics of individual sources (Section 3.1) classified in Section
2.4. An overview of INPs observed at Mt. Helmos is presented as a function of $T$ and the results are contrasted against literature
observations, for both global and specifically for the Mediterranean region (Section 3.2). The INP abundance and its correlation
with aerosol properties of each INP source are then examined. We also focus on a case study where precipitation effects on
INPs in the PBL are explicitly studied (Section 3.3). Finally, the ability of published parameterizations to reproduce observed
INPs at Mt. Helmos is examined, followed by the introduction of new parameterizations we develop that explicitly consider
characteristics of different INP sources and display superior performance (Section 3.4).

### 3.1 Properties of identified aerosol sources

### 3.1.1 The particle size distribution for different aerosol types and airmass classifications

Figure 4a shows the combined particle size distribution of different INP sources measured by both SMPS and APS. Scatter
plots in Fig. 4b and 4c present the apportionment of fine (<1.0 μm, $Fine_{SMPS+APS}$ in Fig. 4b and $Fine_{APS}$ in Fig. 4c) and coarse
(>1.0 μm, $Coarse_{APS}$) particles for different aerosol sources. The results of the concentration of aerosol particles in different
size ranges are provided in Fig. S2. Based on all observations, we summarize that aerosols at $(HAC)^2$ during CALISHTO
generally exhibit four size modes: an ultrafine mode ($D_a$<0.04 μm), an Aitken mode (0.04<$D_a$<0.1 μm), an accumulation mode
(0.1<$D_a$<1.0 μm), and a coarse mode ($D_a$>1.0 μm). When $(HAC)^2$ is in the FT, aerosol particles in the size range $D_a$>0.1 μm
(Fig. 4a), with and without the influence of precipitation/clouds, exhibit a similar size distribution with characteristic decrease
of number with increasing size resembling a power-law (Kim et al., 1992). Particles larger than 4.0 μm in the FT show
negligible amounts. Precipitation/clouds decrease the amount of particles smaller than 0.4 μm, while the absence of a distinct
accumulation mode may suggest inappreciable influences from anthropogenic surface emissions on FT aerosols (Ran et al.,
2022).



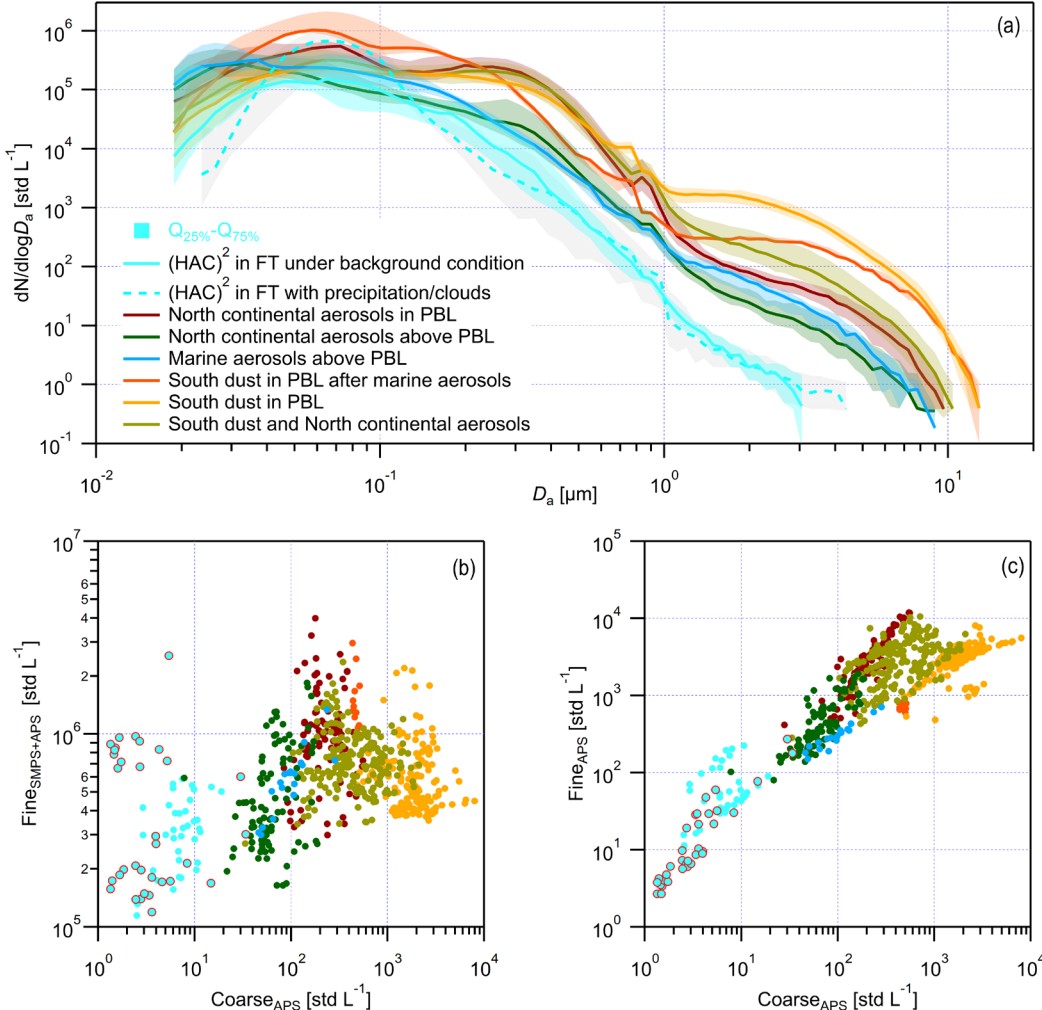

**Figure 4. Particle size distribution of different aerosol sources. (a) Combined size distribution of particles measured by SMPS (10–800 nm in mobility diameter) and APS (0.5−20 μm in aerodynamic diameter). The solid line is the median of the particle size distribution. The shading area shows the 25th to 75th percentiles of the particle size distribution of each aerosol source. (b) Scatter plots of Coarse$_{APS}$ particle (>1.0 μm, on the $x$-axis) number concentration and Fine$_{SMPS+APS}$ particle (particles <1.0 μm measured by both SMPS and APS, on the $y$-axis) number concentration. (c) Scatter plots of Coarse$_{APS}$ particle (>1.0 μm, on the $x$-axis) number concentration and Fine$_{APS}$ particle (particles <1.0 μm measured by APS, on the $y$-axis) number concentration. The aerosol sources are indicated in the legend in panel a.**

FT aerosols with $D_a$<0.1 μm under both background and precipitation/clouds conditions show a broad Aitken mode, but the latter condition tends to supress the ultrafine and accumulation modes (Fig. 4a). Such a size growth for Aitken mode particles and a number concentration decrease in ultrafine particles may be facilitated by the larger temperature and relative humidity gradients during precipitation/clouds periods both of which promote particle growth and gas-to-particle conversion processes (Schroder and Strom, 1997; Bates et al., 1998; Kamra et al., 2003; Peter et al., 2010). Such processes are also promoted by particle decreases in the accumulation mode, leading to decreased sinks for ultrafine particles and gas phase materials (Khadir





et al., 2023). As shown in Fig. 4b, the increase in Fine$_{SMPS+APS}$ particles may not always happen in the FT with the presence of precipitation/clouds. Nonetheless, while new particle formation and particle growth are often reported under PBL conditions, we show that it may also occur under FT conditions (Kerminen et al., 2018).

Among all aerosol sources, North continental aerosols in PBL shows the largest particle concentration in the accumulation mode indicating substantial influences from anthropogenic particles (Zaizen et al., 2004). North continental aerosols in PBL also contains some Aitken and coarse mode particles. Above the PBL, an overall particle number decrease for Aitken, accumulation and coarse modes can be observed for North continental aerosols while it shows an increase in ultrafine mode particles, which is in agreement with Ristovski et al. (2010) reporting the size distribution of continental aerosols penetrated

into the FT. Showing an overall overlapped Q$_{25\%}$–Q$_{75\%}$ range (Fig. 4a), marine aerosols have a similar particle size distribution as North continental aerosols when both are above the PBL. Following marine aerosols, a dust plume intrusion period in the PBL (South dust in PBL after marine aerosols) shows a particle size distribution with substantial increases in coarse, accumulation and Aitken modes, which agrees with the aerosol content increase presented in Fig. 2c compared to Fig. 2b, but it shows a slight decrease in ultrafine mode particles. The large increase in coarse mode particles, especially particles larger

than 10 μm (Fig. 4a), suggests a concurrent presence of dust particles (Brunner et al., 2021). The increase in accumulation and Aitken mode particles may be attributed to the diurnal cycling of (HAC)$^2$ in and out of the PBL. Gini et al. (2022) noted that Saharan dust events are usually associated with lower PBLH. The distinct size distribution characteristic of South dust in PBL is its highest concentration of coarse mode particles among all aerosol sources but the source contains less fine particles (<1.0 μm, Fig. 4b and c). Lastly, the source of South dust and North continental aerosols shows a size distribution with coarse mode

particles between South dust in PBL and North continental aerosols in PBL, and the mixed source shows slightly less particles in the other modes compared to North continental aerosol source in PBL.

**3.1.2 Fluorescent properties of particles from different aerosol sources**

Figures 5 and 6 present the relative fractions and size distributions of different types of fluorescent particles for different aerosols sources, respectively. Figure 7 provides the results of Fluo$_{WIBS}$ particle number concentration, asphericity factor of

Fluo$_{WIBS}$ particles, the correlation between Fluo$_{WIBS}$ particles and dust (eBC) mass concentration in different aerosol sources. Figure 5a shows that when (HAC)$^2$ is in the FT under background condition without the influence of remotely transported airmasses, fluorescent particles are mainly comprised of B$_{WIBS}$, C$_{WIBS}$ and BC$_{WIBS}$ types, as well as a few A$_{WIBS}$ and ABC$_{WIBS}$ particles. The particles detected in B$_{WIBS}$, C$_{WIBS}$ and BC$_{WIBS}$ channels may be attributed to non-biological fluorescent particles (Crawford et al., 2016; Ziemba et al., 2016), whereas the signals for A$_{WIBS}$ and ABC$_{WIBS}$ channels are probably from bioaerosol

(Crawford et al., 2015). Despite a low number concentration of fluorescent particles when (HAC)$^2$ is in the FT under background condition (Figs. 6 and 7a), some of the fluorescent particles with biogenic materials may contribute to the INPs. With the influence of precipitation/clouds in the FT, the dominance of A$_{WIBS}$ and ABC$_{WIBS}$ particles and the fraction increase in AB$_{WIBS}$ particles (Fig. 5a) may be attributed to bacteria for small-sized particles (<2.0 μm) and fungal spores or plant fragments for large-sized particles (>2.0 μm). This is consistent with the release of biological particles during or after



precipitation/clouds events (Prenni et al., 2013; Joung et al., 2017; Iwata et al., 2019; Negron et al., 2020; Khadir et al., 2023).

Those biological particles are potentially active INPs at warm temperatures.

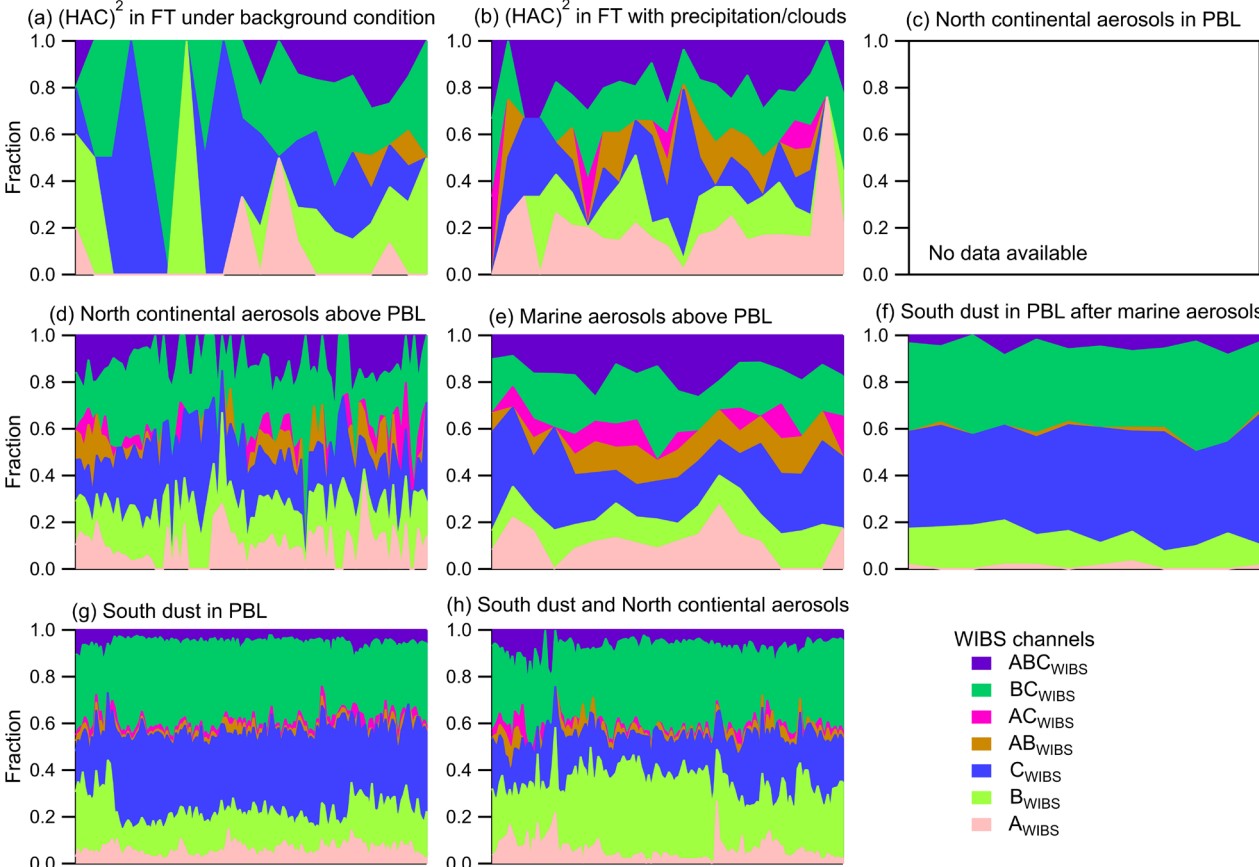

**Figure 5. The fraction pattern of different types of fluorescent particles classified by WIBS. Different particle types are indicated in the legend. The fraction of each type of particle from an identified aerosol source is shown by the *x*-axis. The *y*-axis scales the observations for the case. WIBS data was resampled every 15 minutes for the source of South dust in PBL after marine aerosols due to a short period of observation (< 3 hours), and hourly averaged results were presented for the rest aerosol sources.**

For North continental aerosols and marine aerosols above the PBL, both aerosol sources contain particles from all WIBS channels and the only difference is that the former has more observation data (Fig. 5d and e). This means both aerosol sources contain similar types of fluorescent particles which might be in a mixed state of biological particles and interfering particles, such as dust particles carrying biological matters. Tang et al. (2022a) reported that biological particles in continental aerosols are majorly observed in $A_{WIBS}$, $AB_{WIBS}$ and $ABC_{WIBS}$ channels. It is also reported that biological particles in marine aerosols may show fluorescence in $A_{WIBS}$, $B_{WIBS}$, $AB_{WIBS}$ and $ABC_{WIBS}$ channels (Kawana et al., 2021; Moallemi et al., 2021). Some of $C_{WIBS}$ (~20%) and $BC_{WIBS}$ (~20%) particles in marine and continental aerosols above PBL may be attributed to residual particles associated with fluorescent materials but not necessarily to be biological particles (Pöhlker et al., 2012).





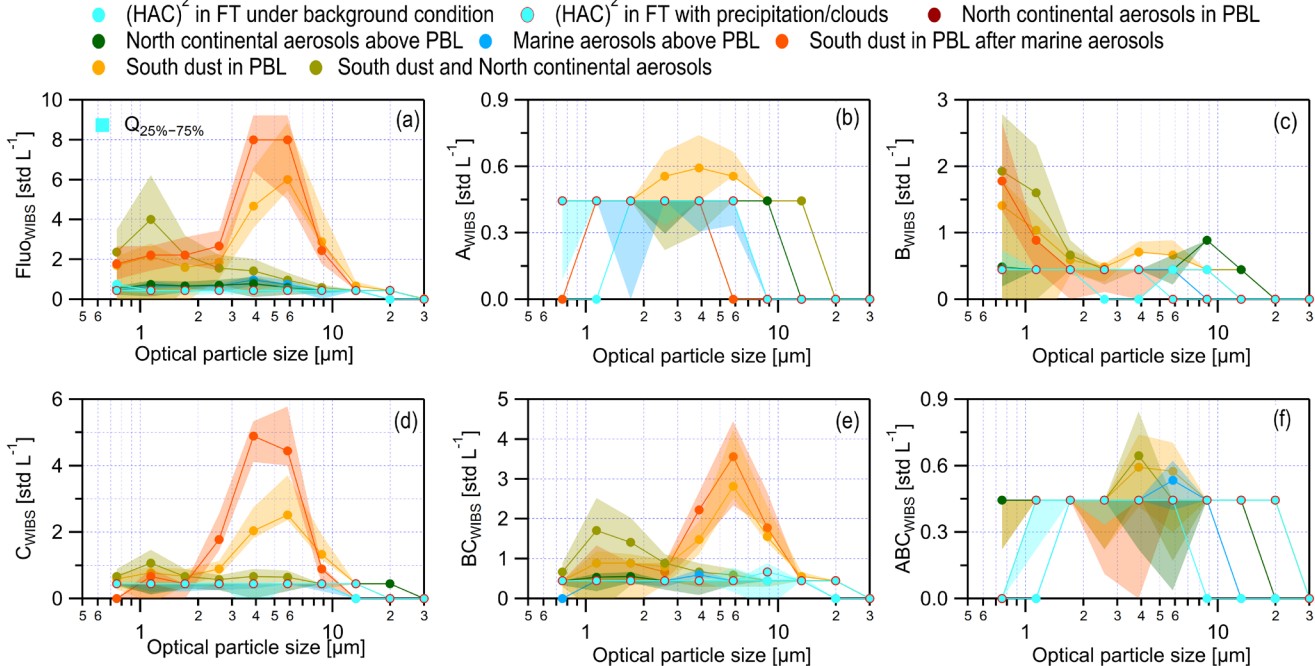

**Figure 6. Fluorescent particle number concentration in 10 different size bins, including [0.50 0.75 µm], (0.75 1.13 µm], (1.13 1.71 µm], (1.71 2.57 µm], (2.57 3.87 µm], (3.87 5.83 µm], (5.83 8.78 µm], (8.78 13.22 µm], (13.22 19.92 µm], (19.92 30.00 µm]. Data points stand for median values. The uncertainty is the range between 25th and 75th percentiles, indicated by $Q_{25\%-75\%}$. Different aerosol sources are indicated in the legend. No WIBS data are available for the source of North continental aerosols in PBL. Different types of fluorescent particles are presented in different panels, including (a) $Fluo_{WIBS}$, (b) $A_{WIBS}$, (c) $B_{WIBS}$, (d) $C_{WIBS}$, (e) $BC_{WIBS}$, (f) $ABC_{WIBS}$. The results for $AB_{WIBS}$ and $AC_{WIBS}$ are not presented because they are minor in all aerosols sources.**

The major types of fluorescent particles for aerosol sources containing dust particles, i.e., South dust in PBL after marine aerosols and South dust in PBL, are $B_{WIBS}$, $C_{WIBS}$ and $BC_{WIBS}$ (Fig. 5). This is consistent with Longo et al. (2014) and Violaki et al. (2021), both of which showed a strong association of bioaerosol with mineral dust plumes. This is also in agreement with Yue et al. (2022) who reported the correlation between dust particles and $C_{WIBS}$ (also $BC_{WIBS}$) particles. It is also reported that soil dust may be co-emitted with organics (O'sullivan et al., 2014) that may show fluorescent in $B_{WIBS}$ and $C_{WIBS}$ channels (Després et al., 2012; Graber and Rudich, 2012). According to the results shown in Fig. 6c, d and e, $B_{WIBS}$ particles in dust-containing aerosol are generally of small sizes (< 2.0 µm) whereas both $C_{WIBS}$ and $BC_{WIBS}$ particles have larger sizes (> 2.0 µm). This suggests that $B_{WIBS}$ particles might be attributed to small-sized soil dust particles, and both $C_{WIBS}$ and $BC_{WIBS}$ particles may be more relevant for large-sized mineral dust particles that show fluorescence. Furthermore, the results in Fig. 7c and d indicate that fluorescent particles in dust plumes may not be purely bioaerosols given the weak correlation between $ABC_{WIBS}$ and dust mass concentration estimated by the SKIRON model ($R = 0.14$ and $\rho = 0.35$). We note that $C_{WIBS}$ and $BC_{WIBS}$ particles might be the most relevant types of fluorescent particles for mineral dust from Saharan dust, supported by the strong and significant correlations between the calculated dust mass concentration and $C_{WIBS}$ (or $BC_{WIBS}$) particles, compared to the other types of fluorescent particles showing insignificant correlations (Fig. S3 in Supplement S3).





**Figure 7. Fluorescent properties of aerosol particles from different sources. (a) Number concentration of Fluo$_{WIBS}$ particles. (b) Asphericity factor of Fluo$_{WIBS}$ particles. (c) Scatter plots of Fluo$_{WIBS}$ particle concentration and dust mass concentration estimated by the SKIRON model at 2170 m a.s.l. (~ 140 m lower than (HAC)$^2$) (d) Scatter plots of ABC$_{WIBS}$ particle concentration and dust mass concentration estimated by SKIRON at 2170 m a.s.l. (e) Scatter plots of Fluo$_{WIBS}$ particle concentration and eBC mass concentration. (f) Scatter plots of ABC$_{WIBS}$ particle concentration and eBC mass concentration. The Pearson correlation coefficient (R) and corresponding p value calculated from F-test, and Spearman's rank coefficient (ρ), are provided to evaluate the correlation between parameters on the x-axis and y-axis.**

Small-sized carbonaceous (Bond et al., 2013) and combustion-generated particles can carry PAHs or other compounds that
fluoresce (Fennelly et al., 2017), and may contribute to the B$_{WIBS}$, C$_{WIBS}$ and BC$_{WIBS}$ populations (Fig. 6c, d and e). This is
supported by the significant correlation ($p < 0.05$) between Fluo$_{WIBS}$ particles and eBC mass concentration (Fig. 7e). ABC$_{WIBS}$
however does not correlate with eBC mass concentration (Fig. 7f) suggesting that ABC$_{WIBS}$ is indeed a distinct population –
biological particles – and not affected by sources that contribute to eBC. Results in Fig. S4 (Supplement S3) show that eBC
particles are generally correlated with B$_{WIBS}$, C$_{WIBS}$ and BC$_{WIBS}$ populations. Compared to the aerosol source of South dust in
PBL, the mixed source of South dust and North continental aerosols shows a larger fraction in B$_{WIBS}$ but a smaller fraction in
C$_{WIBS}$ particles (Fig. 5g and h). The higher content of B$_{WIBS}$ particles may be explained by a larger particle number
concentration of small-sized carbonaceous particles (< 2.0 μm), such as soot from anthropogenic emissions in continental





sources. Furthermore, the small fraction of A$_{WIBS}$ and ABC$_{WIBS}$ particles in dust (Fig. 5g and h) suggests the presence of biological particles from near ground sources in the PBL. This can be explained by the high PBLH conditions (Fig. 3a) that

exceed the (HAC)$^2$ altitude, meaning that the site is in the PBL and directly influenced by bioaerosols emitted by the surrounding forested area, such as bacteria, fungi and/or fungal spores.

Comparing Fluo$_{WIBS}$ and ABC$_{WIBS}$ particles in different aerosol sources in and above the PBL (Figs. 6 and 7), it shows that sources in the PBL contain more fluorescent particles by approximately one order of magnitude than those sources above the PBL. This suggests particles from biogenic sources can be significantly reduced when the atmospheric condition changes from

the PBL to FT. In addition, we discuss the impacts of anthropogenic emissions on different aerosol sources by using the observed eBC results. Figure 7e and f show that all identified aerosol sources contain eBC, despite low eBC concentrations under conditions in the FT with and without precipitation/clouds (<0.05 μg std m$^{-3}$). This means anthropogenic pollution is possible to impact (HAC)$^2$ even when it is in the FT (Collaud Coen et al., 2018). Also, it shows that the occurrence of precipitation/clouds does not completely remove eBC (Fig. 7e and f), likely due to its hydrophobic properties (Gao et al.,

2022). North continental aerosols show a higher eBC concentration in the PBL than above PBL, also indicating the importance of atmospheric condition in determining anthropogenic emission impacts. The presence of eBC in long range transported marine aerosols above the PBL may suggest the long atmospheric residence time of eBC. The high eBC mass concentration observed in South dust aerosols may be due to the air mass exchange between dust plume and PBL aerosols, given that dust events may suppress PBL air masses and enhance air parcel entrainment (Zhang et al., 2022).

**3.2 INPs at Mt. Helmos**

**3.2.1 Overview**

INP concentrations at Mt. Helmos (Fig. 8) generally increase with decreasing $T$; the concentration spans from 10$^{-3}$ to 10$^2$ std L$^{-1}$. For $T > -15$ °C, the observed INPs may be attributed to biological particles, especially at the warmest temperatures (Murray et al., 2012; Kanji et al., 2017). These bioaerosols may originate from local and/or transported continental biogenic

sources, such as vegetation and living organisms, and also from sea surface microlayer which releases marine diatoms and/or diatom exudates (Després et al., 2012). Soil dust emitted from agricultural lands (local and continental), which are reported to be able to nucleate ice at $T > -10$ °C (Garcia et al., 2012; Harrison et al., 2016; Hill et al., 2016), and organic material emitted by marine organisms, some of which can freeze at $T$ close to $-10$ °C via the immersion freezing mode (Wilson et al., 2015), may also contribute to INPs observed at warm temperatures. The higher median INP concentrations at $T > -10$ °C observed at

Mt. Helmos compared to those of lower temperatures (at $-11$ °C and $-12$ °C) are due to the limited observations at warm temperatures. Also, we note that the INSEKT measurement uncertainty is higher at warmer $T$. For $T < -15$ °C ($> -27$ °C), mineral dust and soil dust of arid and agricultural origins, are more important INPs than bioaerosols (Hoose and Möhler, 2012; Murray et al., 2012; Tobo et al., 2014). Moreover, local and transported ash particles emitted from industrial coal combustion and domestic fuel use, may contribute to the observed INPs for $T < -15$ °C (Umo et al., 2015). Additionally, we note that INPs



observed by PINE at $T = -24$ °C show higher concentrations than those measured by INSEKT at the same $T$. This may partly result from the decreased particle collection efficiency of INSEKT filters due to over loading under high particle concentration conditions. The INP concentration difference is more pronounced at $T = -24$ °C because this temperature was employed for PINE experiments to observe a long period dust event from November 4 to 10. More details about the difference between INPs recorded by PINE and INSEKT are provided in the Supplement S4 (Figs. S5 and S6). Note that PINE, measuring INPs

activated by different IN mechanisms (Möhler et al., 2021), is technically different from INSEKT that only detects INPs in the immersion freezing mode. This may also partly explain the higher INP concentrations observed from PINE compared to INSEKT. Moreover, some INPs may not be tested by INSEKT if aerosol particles collected on the filter are not extracted completely. Therefore, PINE reports higher INP concentrations than INSEKT because PINE tests INPs in total aerosols.

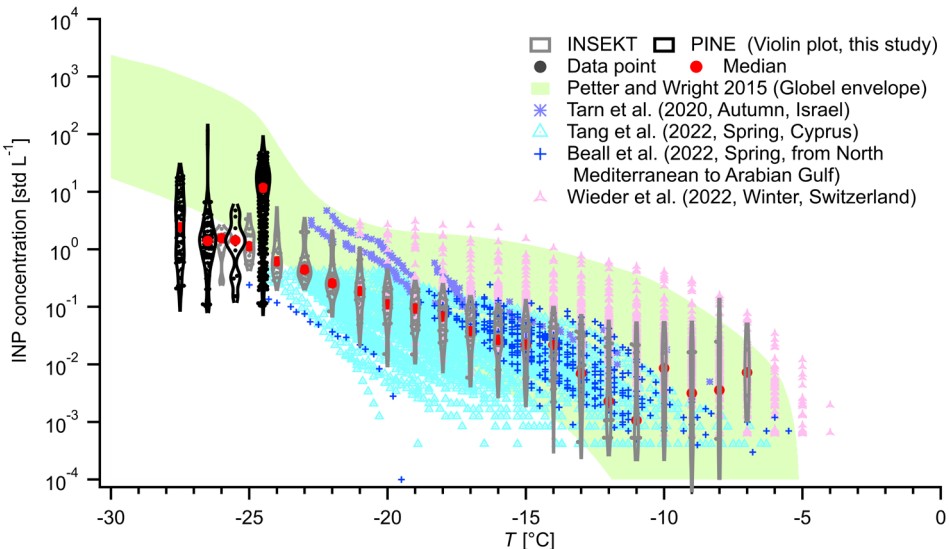

**Figure 8. INP concentrations observed at Mt. Helmos as a function of temperature ($T$) contrasted against INP levels reported in selected field campaigns in Mediterranean region (Tarn et al., 2020; Beall et al., 2022; Tang et al., 2022b; Wieder et al., 2022) and from an global envelope (Petters and Wright, 2015). INP data measured by offline INSEKT and online PINE in this study are presented as violin plots, indicated by grey and black violin box, respectively. Each violin box presents data points in a $T$ interval of 1 °C. Red round markers in the violin box represent the median value. Symbols for PINE INPs are offset by −0.5 °C for legibility.**

Figure 8 also shows that INPs at Mt. Helmos are generally distributed in the global envelope range reported by Petters and Wright (2015), particularly for $T > -25$ °C, and are also consistent with the results reported for other campaigns in Mediterranean region (Tarn et al., 2020; Beall et al., 2022; Tang et al., 2022b). Wieder et al. (2022) observed higher INP concentrations over Alps in wintertime compared to this study for $T > -20$ °C. This may be attributed to a larger vegetation coverage (or differences in species, rainfall or season) in the wintertime Alps, hence larger bioaerosol sources relevant for

INPs, compared to Mt. Helmos. Also notable is that Wieder et al. (2022) report INPs active at $T$ as high as −4 °C, which implies the presence of INPs that are more efficient in the Alps than at Mt. Helmos.

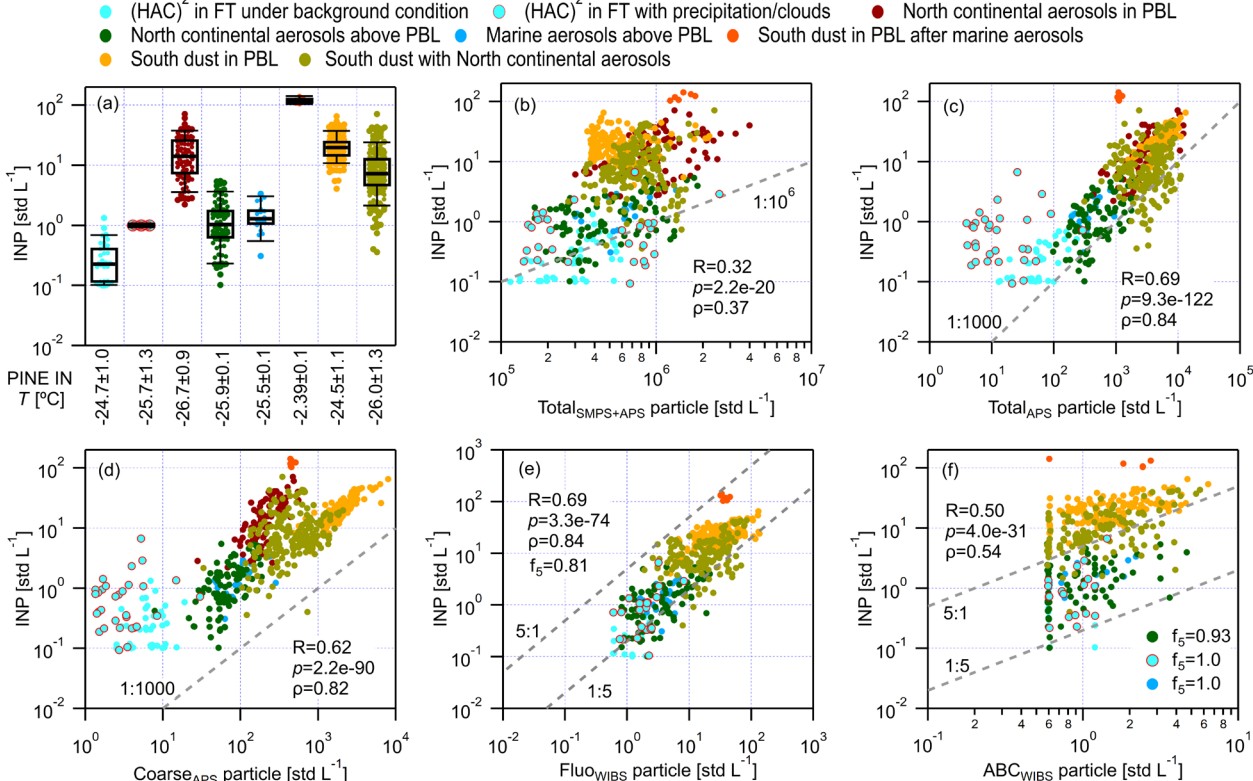

**Figure 9. PINE INP concentration in different aerosol sources and the relation between INP and aerosol particle concentrations. (a) Box plots for INP concentration in different aerosol sources. The average $T$ for PINE IN experiments for each source is indicated at the bottom axis and the uncertainty stands for one standard deviation. (b) Scatter plots of INP and Total$_{SMPS+APS}$ concentrations. (c) Scatter plots of INP and Total$_{APS}$ concentrations. (d) Scatter plots of INP and Coarse$_{APS}$ concentrations. (e) Scatter plots of INP and Fluo$_{WIBS}$ concentrations. (f) Scatter plots of INP and ABC$_{WIBS}$ concentrations. The Pearson correlation coefficient ($R$) and corresponding $p$ value calculated from F-test, and Spearman's rank coefficient ($\rho$), are provided to evaluate the correlation between INP concentration and different aerosol particle concentrations. The $p$ value is the probability of obtaining an $R$ value no smaller than the true $R$ value if there is no linear correlation between INPs and the given parameter.**

### 3.2.2 INP concentrations for different sources

The high frequency of PINE measurements allows for the calculation of hourly-averaged INP concentrations for different aerosol sources. The small $T$ spread (< 4.0 °C) of PINE measurement conditions throughout the campaign also allows for a thorough comparison of INP abundance from different sources (at similar $T$). As shown in Fig. 9, INP concentration is less than 1.0 std L$^{-1}$ ($T = -24.7$ °C) when (HAC)$^2$ is in the FT under background condition without precipitation/clouds. This is in agreement with Lacher et al. (2021) who measured INPs at Jungfraujoch in Switzerland under FT conditions (reported range, 0.01 and 1.0 std L$^{-1}$ at $-25$ °C in immersion freezing mode). With the influence of precipitation/clouds, INPs at (HAC)$^2$ in the FT approximately increase to 1.0 std L$^{-1}$ ($T = -25.7$ °C). The enriched INPs may be attributed to cloud-processed particles (Khadir et al., 2023) and the near-ground released bioaerosols (likely A$_{WIBS}$, AB$_{WIBS}$ and ABC$_{WIBS}$ in Fig. 5a) produced by the precipitation splash (Prenni et al., 2013; Joung et al., 2017). Cloud-processed particles, being a result of evaporated/sublimated





hydrometeors, are originally active INPs and show enhanced IN ability compared to the particles before cloud-processing (Jahl et al., 2021). In addition, during precipitation/clouds periods, PBL airmasses containing bioaerosols close to the cloud base may be entrained into the FT close to the cloud top, such that more INPs are measured at $(HAC)^2$ in the FT. We also note that

the overall 1.0 °C lower $T$ conditions for PINE experiments for $(HAC)^2$ in the FT with precipitation/clouds may also partly contribute to its higher tested INPs.

**Table 2. The Pearson correlation coefficient ($R$) and Spearman's rank coefficient ($\rho$) for the relationship evaluation between INP and aerosol particle concentration from different sources. A critical $p$ value of 0.05 from F-test for $R$ is used to assess the significance level of the relationship. A $p$ value smaller than 0.05 suggests that the probability of obtaining an $R$ value no smaller than the true $R$**
**value is less than 5% if there is actually no liner correlation between INPs and the given parameter, thus the calculated $R$ is of statistical significance. Evaluated significant relationship is indicated in bold. Note that the correlation between INPs and Total$_{WIBS}$ particles is not included in Fig. 9 but provided in Fig. S7 in Supplement S5.**

| INP sources | Total$_{SMPS+APS}$[a] $R$ ($p$) | $\rho$ | Total$_{APS}$[b] $R$ ($p$) | $\rho$ | Coarse$_{APS}$[c] $R$ ($p$) | $\rho$ | Total$_{WIBS}$[d] $R$ ($p$) | $\rho$ | Fluo$_{WIBS}$[e] $R$ ($p$) | $\rho$ | ABC$_{WIBS}$[f] $R$ ($p$) | $\rho$ |
|---|---|---|---|---|---|---|---|---|---|---|---|---|
| $(HAC)^2$ in FT under background condition* | **0.41** **0.01** | 0.57 | **0.76** **7.3e–8** | 0.66 | 0.13 0.46 | 0.19 | 0.41 0.07 | 0.31 | **0.53** **0.03** | 0.34 | −0.50 0.22 | −0.43 |
| $(HAC)^2$ in FT with precipitation/clouds | 0.28 0.10 | 0.06 | 0.09 0.62 | 0.11 | 0.04 0.81 | 0.03 | 0.17 0.45 | 0.27 | **0.52** **0.01** | 0.56 | **0.56** **0.01** | 0.33 |
| North continental aerosols in PBL | **0.33** **0.003** | 0.35 | **0.71** **1.1e–14** | 0.78 | **0.73** **6.1e–16** | 0.72 | NA[g] | NA | NA | NA | NA | NA |
| North continental aerosols above PBL | **0.53** **9.1e–8** | 0.34 | **0.53** **6.4e–8** | 0.48 | **0.59** **1.3e–9** | 0.54 | **0.53** **1.3e–7** | 0.45 | **0.62** **2.2e–10** | 0.58 | **0.32** **0.008** | 0.30 |
| Marine aerosols above PBL | **0.61** **0.007** | 0.48 | **0.53** **0.02** | 0.69 | **0.48** **0.04** | 0.69 | **0.60** **0.009** | 0.67 | **0.55** **0.02** | 0.47 | **0.71** **9.1e–4** | 0.71 |
| South dust in PBL after marine aerosols | 0.59 0.21 | 0.71 | 0.12 0.82 | 0.26 | 0.01 0.99 | 0.03 | −0.17 0.75 | −0.26 | −0.38 0.46 | −0.43 | −0.59 0.29 | 0.36 |
| South dust in PBL | 0.05 0.56 | 0.01 | **0.80** **2.2e–33** | 0.80 | **0.84** **2.3e–39** | 0.85 | **0.84** **2.3e–39** | 0.85 | **0.56** **2.3e–13** | 0.52 | **0.59** **1.3e–14** | 0.50 |
| South dust with North continental aerosols | **0.29** **5.3e–5** | 0.09 | **0.24** **0.001** | 0.21 | 0.13 0.07 | 0.34 | **0.21** **0.01** | 0.29 | **0.59** **2.1e–14** | 0.38 | **0.65** **1.3e–17** | 0.51 |
| All observations | **0.32** **2.2e–20** | 0.37 | **0.69** **9.3e–122** | 0.84 | **0.62** **2.2e–90** | 0.82 | **0.73** **3.3e–87** | 0.89 | **0.69** **3.3e–74** | 0.84 | **0.50** **4.0e–31** | 0.54 |

[a] Total particle (0.01 – 20.0 μm) number concentration measured by both SMPS and APS; [b] Total particle (0.5 – 20.0 μm) number concentration measured by APS; [c] Coarse particle (> 1.0 μm) number concentration measured by APS; [d] Total particle (0.5 – 30.0 μm in

optical size) number concentration measured by WIBS; [e] Number concentration of particles showing fluoresce in any one of WIBS fluorescent channels; [d] Number concentration of particles showing fluoresce in all three WIBS fluorescent channels; [g] Data not available

The concentration of INPs from North continental aerosols in PBL is between 2.2 and 71.6 std L$^{-1}$, showing a median of 14.1 std L$^{-1}$ ($T = -26.7$ °C). When North continental aerosols serve as INP sources for $(HAC)^2$ above PBL, the observed INP concentration decreases substantially (about tenfold). The results suggest that the INP concentration from a similar continental air mass depends on its relative position in the atmosphere and is consistent with published studies to date (e.g., Gong et al.

(2022)). INPs in marine aerosols above the PBL range between 0.3 and 3.4 std L$^{-1}$ (median 1.3 std L$^{-1}$) at $T = -25.5$ °C, which is higher than the values reported in Lacher et al. (2021) from a source attributed to marine aerosols from western Mediterranean Sea. It is likely that the lower INPs (even at a lower experiment $T$) observed by Lacher et al. (2021) at Jungfraujoch in Switzerland are a result of longer transportation of marine aerosols, during which particle deposition and



aging-induced deactivation may occur (Schrod et al., 2020). The dust plume after marine aerosols shows the highest INP concentration among all sources, showing an average of $121.3\pm14.7$ std $L^{-1}$ and a median of 121.1 std $L^{-1}$ ($T = -23.9$ °C, resampled for every 15 min due to short period of observations < 3 hours). Such a high INP concentration in Saharan dust was also observed at Jungfraujoch in Switzerland (>200 std $L^{-1}$ for $T = -30$ °C) (Brunner et al., 2021). Also, the observed INP concentration is substantially higher than $ABC_{WIBS}$ concentration (<10 std $L^{-1}$ in Fig. 9f), suggesting that dust particles – but

not any associated biological particles – make the primary contribution to the observed INPs in the dust plume. The median INP concentration decreases to 19.8 # std $L^{-1}$ ($T = -24.5$ °C) when dust plume is more extensively mixed with local aerosols in the PBL, i.e., the source of South dust in PBL. Furthermore, the median INP concentration decreases further to 7.3 # std $L^{-1}$ ($T = -26.0$ °C) when the dust plume is also mixed with continental aerosols, i.e., the source of South dust with North continental aerosols. It is likely that the source of South dust in PBL after marine aerosols may contain more fresh dust particles than the

following sources having more aged and deactivated dust particles (Boose et al., 2019).

Among all sources presented in Fig. 9, INP concentration in the PBL is considerably larger than that in the FT by approximately more than one order of magnitude (median value), although it depends on INP sources. Both continental aerosols from the North and dust from the South are major sources for INPs at Mt. Helmos. Fresh dust plume contains a larger number of INPs than the other sources mixed with local emissions and/or continental aerosols. Such a decrease in INP abundance in the mixed

dust containing sources may result from dilution of air masses or aerosol aging induced INP deactivation. Also, we note that the INP concentration range in North continental aerosols ($T = -26.7$ °C) is analogous to that of South dust ($T = -24.5$ °C) when both are mixed with local emissions in the PBL.

Figure 9 and Table 2 also provide the correlation between INP concentration and different concentrations of aerosol particles, including $Total_{SMPS+APS}$, $Total_{APS}$, $Coarse_{APS}$, $Fluo_{WIBS}$ and $ABC_{WIBS}$. The concentration of PINE INPs is approximately higher

than 1 per $10^6$ of $Total_{SMPS+APS}$, 1 per $10^3$ of $Total_{APS}$ and 1 per 500 of $Coarse_{APS}$, respectively, consistent with the established view that INPs in the atmosphere show a size dependence and that larger-sized aerosol particles are of higher probability of behaving as INPs. Overall, a significant and positive correlation between INP concentration and aerosol particle concentrations can be found in Fig. 9. A $\rho$ value larger than 0.80 for the relation between INPs and $Total_{APS}$, or $Coarse_{APS}$, or $Fluo_{WIBS}$ (>0.5 μm, optical size) means that INP concentration increases with those three particle concentrations following a strong monotonic

trend. In comparison to $Total_{APS}$, $Coarse_{APS}$, $Fluo_{WIBS}$ (Fig. 9) or $Total_{WIBS}$ (Fig. S7), the smaller $R$ and $\rho$ values for the relation between $Total_{SMPS+APS}$ and INPs also indicates that small-sized aerosol particles in the SMPS size range may play a secondary role for serving as INPs compared to larger-sized particles measured in the APS and WIBS size range. The IN dependence on the size of aerosol particles is more pronounced for the sources of North continental aerosols and South dust when both sources supply potential INPs for $(HAC)^2$ in the PBL, as shown that the $R$ value between INPs and $Total_{APS}$ or $Coarse_{APS}$ for South

dust in PBL does not show significant difference compared to the R value between INPs and $Total_{SMPS+APS}$ (Table 2). This is because the source of South dust in PBL contains a much smaller proportion of fine particles to the total than for the source of North continental aerosols in PBL (Fig. 4). Also, the $\rho$ value of the relation between INPs and $Coarse_{APS}$ particles for North





continental aerosols is larger than that of South dust in PBL, suggesting that INPs in North continental aerosols may be more dependent on coarse mode particles.

In addition to size dependence, we note that the observed INP concentration is close to the concentration of fluorescent particles, as shown that more than 80% of INP data points are within a factor of 5 compared to $Fluo_{WIBS}$ data (Fig. 9e). Also, INPs from most sources show a significant correlation with $Fluo_{WIBS}$ particles (except for the source of South dust in PBL after marine aerosols owing to limited number of observations). This suggests that particles showing fluorescence are of significant relevance for INPs observed at Mt. Helmos. The results are also in agreement with Mason et al. (2015) who reported that INPs observed between −15 and −25 °C at a coastal site are strongly correlated with fluorescent particles. $ABC_{WIBS}$ would constitute a subset of the observed total INPs, as the latter is approximately five times larger (Fig. 9f) and shows a significant correlation with $ABC_{WIBS}$ particles in the source (Table 2). In particular, more than 90% observed INP data from sources of aerosols in the FT influenced by precipitation/clouds, marine aerosols and continental aerosols above the PBL, are within a range less than a factor of 5 compared with $ABC_{WIBS}$ particles in the source (Fig. 9f). Such a close correlation highlights the importance of biological particles in those INP sources when dust particles are absent. Notably, the correlation between INPs and $ABC_{WIBS}$ particles for aerosols in the FT influenced by precipitation/clouds becomes significant compared to the case without precipitation/clouds effects (Table 2), suggesting that precipitation/clouds may lead to an increase in $ABC_{WIBS}$ (Fig. 7) and contribute to observed INPs. Lastly, Table 2 shows that of all sources, $ABC_{WIBS}$ particles from marine aerosols show the strongest correlation with INPs. This is consistent with the important role of marine biogenic aerosols in serving as INPs in the MPC regime (Wilson et al. (2015).

### 3.2.3 The ice nucleation ability of particles in different aerosol sources

Figure 10a uses the ratio of INP concentration to $Total_{SMPS+APS}$ concentration to estimate the INP proportion in total aerosol particles for different sources and uses the ratio as a measure to evaluate the average IN ability of aerosol particles from different sources. To clarify, we note that the ratio statistically refers to the overall ice formation ability of the particle population in the source. However, the ratio is not relevant to the IN ability of single particles which specifically relies on the physiochemical properties of the particle, given that sources containing low concentration of INPs may have effective INPs activating at warm temperatures. When $(HAC)^2$ is in the FT under background condition without remotely transported air masses and without precipitation/clouds, the observed INP ratio is approximately one per million aerosol particles and the median ratio value is less than that reported in Rogers et al. (1998) who reported ~30 INPs out of a million particles at an altitude of 10.6 km in the upper troposphere and in a *T* range between −15 and −40 °C. Influenced by precipitation/clouds, the INP ratio in the FT generally increases because of the decrease in total aerosol particles and the increase in INPs (Section 3.1 and 3.2.2). Figure 10a also suggests that $(HAC)^2$ position *wrt.* PBL regulates the average IN ability of particles from the North continental aerosol source, shown a larger INP ratio when the source is in the PBL than above the PBL. This is because active INPs from the source in the PBL may majorly come from $Coarse_{APS}$ particles which otherwise take a smaller proportion when the source is above the PBL (Fig. 10b). The INP ratio of marine aerosols above PBL is analogous to that of North continental





aerosols above PBL (Fig. 10a), suggesting a similar IN ability of particle populations in both aerosol sources. In addition, the INP ratio of aerosol sources containing dust particles decreases if the source is more influenced by the PBL or if it is mixed with North continental aerosols.

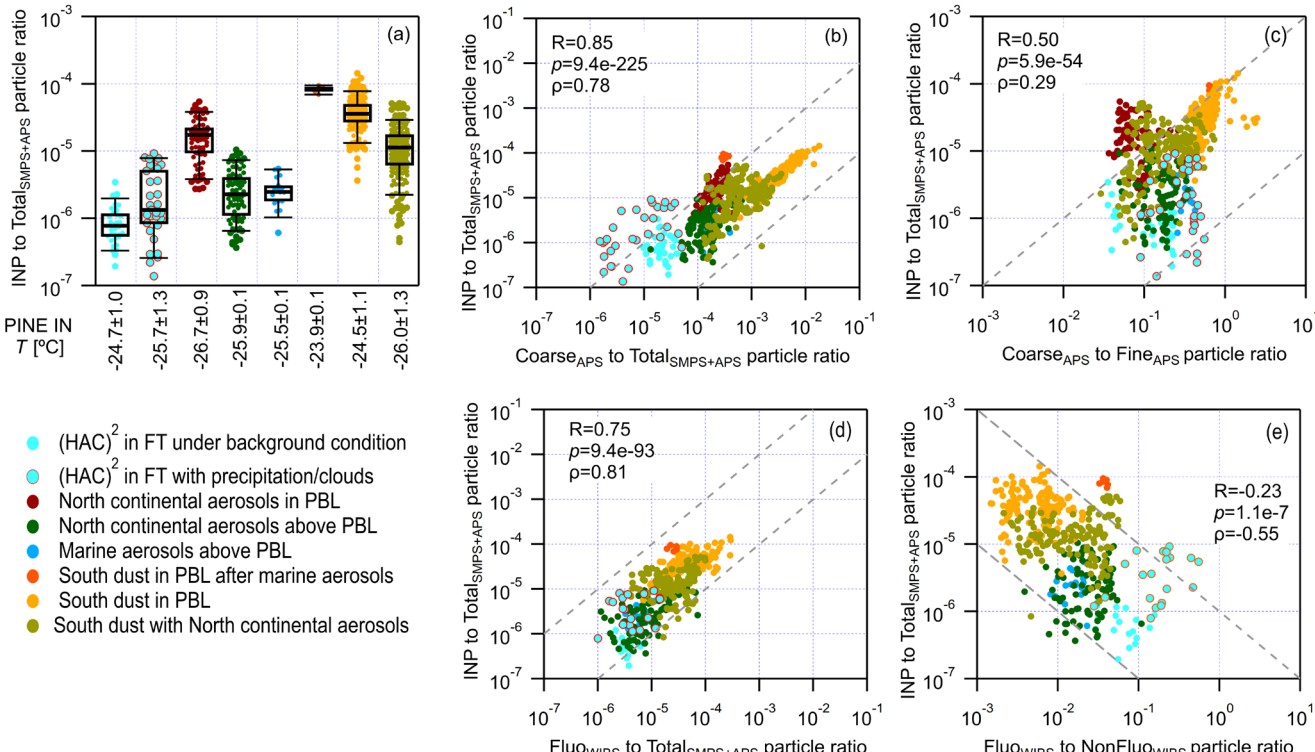

**Figure 10. (a) Box plots for the ratio of INP concentration to Total$_{SMPS+APS}$ concentration. The average $T$ for PINE IN experiments for each INP source is indicated at the bottom axis and the uncertainty stands for one standard deviation. (b) Scatter plots of the ratio of INPs to Total$_{SMPS+APS}$ particles and the ratio of Coarse$_{APS}$ to Total$_{SMPS+APS}$ particles. (c) Scatter plots of the ratio of INPs to Total$_{SMPS+APS}$ particles and the ratio of Coarse$_{APS}$ to Fine$_{APS}$ particles. (d) Scatter plots the ratio of INPs to Total$_{SMPS+APS}$ particles and the ratio of Fluo$_{WIBS}$ to Total$_{SMPS+APS}$ particles. (e) Scatter plots of the ratio of INPs to Total$_{SMPS+APS}$ particles and the ratio of Fluo$_{WIBS}$ to NonFluo$_{WIBS}$ (the difference between Total$_{WIBS}$ and Fluo$_{WIBS}$) particles. The Pearson correlation coefficient ($R$) and corresponding $p$ value calculated from F-test, and Spearman's rank coefficient ($\rho$), are provided to evaluate the correlation between INP abundance and the particle partitioning. The $P$ value is the probability of obtaining an $R$ value no smaller than the true $R$ value if there is no liner correlation between INPs and the given parameter. The grey dashed lines in the panel confine a range of two magnitude on both $x$–axis and $y$–axis.**

The results in Fig. 10 also evaluate the dependence of INP ratio on the particle size and fluorescent property portioning condition of the source, including the ratio of Coarse$_{APS}$ to Total$_{SMPS+APS}$, Coarse$_{APS}$ to Fine$_{APS}$, Fluo$_{WIBS}$ to Total$_{SMPS+APS}$ and Fluo$_{WIBS}$ to NonFluo$_{WIBS}$ (the difference between Total$_{WIBS}$ and Fluo$_{WIBS}$). In general, the average INP ratio of a source increases with the increasing proportion of Coarse$_{APS}$ particles (> 1.0 μm, Fig. 10b and c) and Fluo$_{WIBS}$ particles (Fig. 10d and e) in the source, but such a correlation varies its strength among individual sources. Figure 10b shows that the INP ratio of North continental aerosols (for both above and in the PBL) has a weaker correlation with Coarse$_{APS}$ particles (see $R$ and $\rho$



values in Table S1 in Supplement S5) compared to that of South dust in PBL. Again, this is because North continental aerosols contain more fine mode particles, which are less effective INPs, than aerosols from South dust in PBL. A larger slope for North continental aerosols in Fig. 10b compared to South dust in PBL suggests that INPs in North continental aerosols are more dependent and sensitive to $Coarse_{APS}$ particles. This may be because individual $Coarse_{APS}$ particles in continental aerosols of

biological origin are generally more effective INPs than those in dusty aerosols. This can be true if those coarse mode particles in continental aerosols are of a biologic origin. Moreover, the insignificant correlation between the INP ratio of sources containing North continental aerosols and the ratio of $Coarse_{APS}$ to $Fine_{APS}$ in the source (Fig. 10c and also Table S1 in Supplement S5) suggests that some particles smaller than the APS size detection range (~0.5 μm) may contribute to the observed INPs at ~ −26 °C. Figure 10c shows that the correlation between INP ratio and $Coarse_{APS}$ to $Fine_{APS}$ ratio is less

significant (see Table S1) for the source of South dust in PBL compared to the correlation between INP ratio and $Coarse_{APS}$ to $Total_{SMPS+APS}$ ratio, also suggesting a contribution of some small-sized particles (<0.5 μm) to observed INPs. This is consistent with a field study in Israel (in the Eastern Mediterranean region) focusing on the IN ability of size-resolved Saharan dust particles and the study reported that 0.3 μm (aerodynamic diameter) dust particles are effective INPs at the $T$ range from −20 °C to −30 °C (Reicher et al., 2019). In addition, the increasing $Fluo_{WIBS}$ to $Total_{SMPS+APS}$ ratio can generally predict the

increasing INP ratio of an aerosol source (Fig. 10d, no data available for the source of North continental aerosols). However, Figure 10e shows that the INP ratio in different aerosol sources overall decreases with increasing $Fluo_{WIBS}$ to $NonFluo_{WIBS}$ ratio. The results on the $x$-axis of Fig. 10e show that the $Fluo_{WIBS}$ to $NonFluo_{WIBS}$ ratios in different sources are in reverse order compared to the other ratios presented in Fig. 10b, c and d. Such a difference suggests the INPs that are of a biological origin become less important when the overall IN ability and INP abundance of the source is higher, such as dust plumes, indicating

the secondary role of biological particles in dust-containing sources in serving as INPs.

Overall, the scatter patterns of INP ratio versus different aerosol partitioning index presented in Fig. 10 spread over two orders of magnitude (confined by grey dashed lines in Fig. 10b, c, d and e). Both $Coarse_{APS}$ to $Fine_{APS}$ ratio and $Fluo_{WIBS}$ to $NonFluo_{WIBS}$ ratio show significant correlations ($p<0.05$ in Table S1) with the ratio of INP to $Total_{SMPS+APS}$ (Fig. 10c and e), although the strengths of these two correlations are weaker compared to the results using $Total_{SMPS+APS}$ data in Fig. 10b and d.

This suggests the proportion of particles with different sizes and fluorescent properties conveys the IN ability of particles from different aerosol sources, which may benefit the prediction of INPs in parameterizations. In Section 3.4, these ratios will be incorporated into INP parameterizations to improve their prediction skill.

## 3.3 The influence of precipitation/clouds on INPs in PBL

### 3.3.1 Different scenarios classified during the case study

In addition to the effects of precipitation/clouds on INPs for periods of $(HAC)^2$ in the FT, precipitation/clouds periods were also observed when the site resided in the PBL – with different aerosol sources however, so we treat them separately. For this we focus on a case study (November 23) where the $(HAC)^2$ is in the PBL and influenced by continental aerosols. The



FLEXPART footprints and the HYSPLIT back trajectories (Figs. S9 and S10, respectively) suggest that North continental airmass dominates the aerosol source at (HAC)$^2$ during this time (with a possible minor contribution of South dust). The
dominance of North continental aerosols is also supported by nephelometer results, as $\alpha \sim 2.0$ (Mordas et al., 2015). Following the methodology introduced in Section 2.4, the observations during the day are classified based on the presence of precipitation/clouds and the position of (HAC)$^2$ *wrt.* PBL, by using Ze and MDV values from radar measurements presented in Fig. 11a and b, respectively. The position of (HAC)$^2$ *wrt.* PBL is evaluated by using SMPS $N_{>95nm}$ time series in Fig. 11c since PBLH results from lidar are not always available during the day. Thus, observations on November 23 are classified into
five periods ("cases"), including (HAC)$^2$ in FT with precipitation (case 1) from 00:00 to 04:00, (HAC)$^2$ in FT with precipitation (case 2) from 05:00 to 09:00, (HAC)$^2$ around PBL and close to the cloud top (case 3) from 10:00 to 15:00, (HAC)$^2$ in PBL (case 4) from 15:30 to 18:00, and (HAC)$^2$ in PBL with precipitation/clouds (basically precipitation below (HAC)$^2$, case 5) from 19:00 to 24:00. The box plots of INP abundance observed by PINE and the correlation between INPs and aerosol particles of different cases are presented in Fig. 12. Additionally, the aerosol property results, including particle size distribution of
different cases (Fig. S11), Total$_{SMPS+APS}$, Coarse$_{APS}$, SMPS $N_{<95\ nm}$, Fluo$_{WIBS}$ and ABC$_{WIBS}$ particle concentration, and eBC mass concentration (Fig. S12), are provided in the Supplement and used to understand the changing INP abundances of different scenarios presented in this section.

### 3.3.2 Particle properties

The aerosol particle properties for each case are shown in Fig. 11c, d and e, and Figs. S11 to S13. Figure 11c shows that PBL
boundary generally evolves from a position below (HAC)$^2$ to a position above (HAC)$^2$ throughout the day, as SMPS $N_{>95nm}$ increases. The increasing PBLH is also indicated by size distribution shifts to coarse-sized particles in Fig. S11 during the day. When (HAC)$^2$ is in the FT with precipitation (case 1), Coarse$_{APS}$ particles at (HAC)$^2$ show a median of ~1.5 std L$^{-1}$ (Fig. S12b). The median value is well below the critical value of 20 std L$^{-1}$ (see Section 2.4), suggesting that the site at this time is exposed to clean background conditions with a low probability of influence from remotely-transported aerosols above the PBL. This is
also supported by the low eBC mass concentration (~0.01 μg std m$^{-3}$) shown in Fig. S12f.

When (HAC)$^2$ is in the FT with precipitation (case 2), Coarse$_{APS}$ particles at (HAC)$^2$ are occasionally larger than 20 std L$^{-1}$ and show a median of ~16.7 std L$^{-1}$ (Fig. S12b). Likely, it suggests that remotely transported continental aerosols exert an influence on aerosols at (HAC)$^2$, although (HAC)$^2$ is still above PBL with SMPS $N_{>95nm}$ smaller than 100 std cm$^{-3}$. Also, it is possible that precipitation from higher altitude clouds compared to case 1 results in downdrafts that drive mass entrainment of
remotely transported aerosols. Furthermore, compared to case 1, the decreased ABC$_{WIBS}$ particles in case 2 suggests negligible biological particles in downdrafts from high altitudes (Fig. S12e) whereas the increased eBC mass concentration is a result of transportation (Fig. S12f). Additionally, the comparison between case 1 and 2 suggests that Coarse$_{APS}$ particle concentration less than 20 std L$^{-1}$ is a more conservative evaluation standard to diagnose (HAC)$^2$ inside or outside the FT compared to the criterion of SMPS $N_{>95nm}$ comparing to 100 std cm$^{-3}$ (also see Section 2.4). In addition to the differences between the vertical
particle sources for case 1 and case 2, we note that the average wind speed decreases from ~13 m s$^{-1}$ (case 1) to 6 m s$^{-1}$ (case




2) (not shown), which would decrease the emission rate of ABC$_{WIBS}$ particles from the near-ground sources, such as soils and trees.

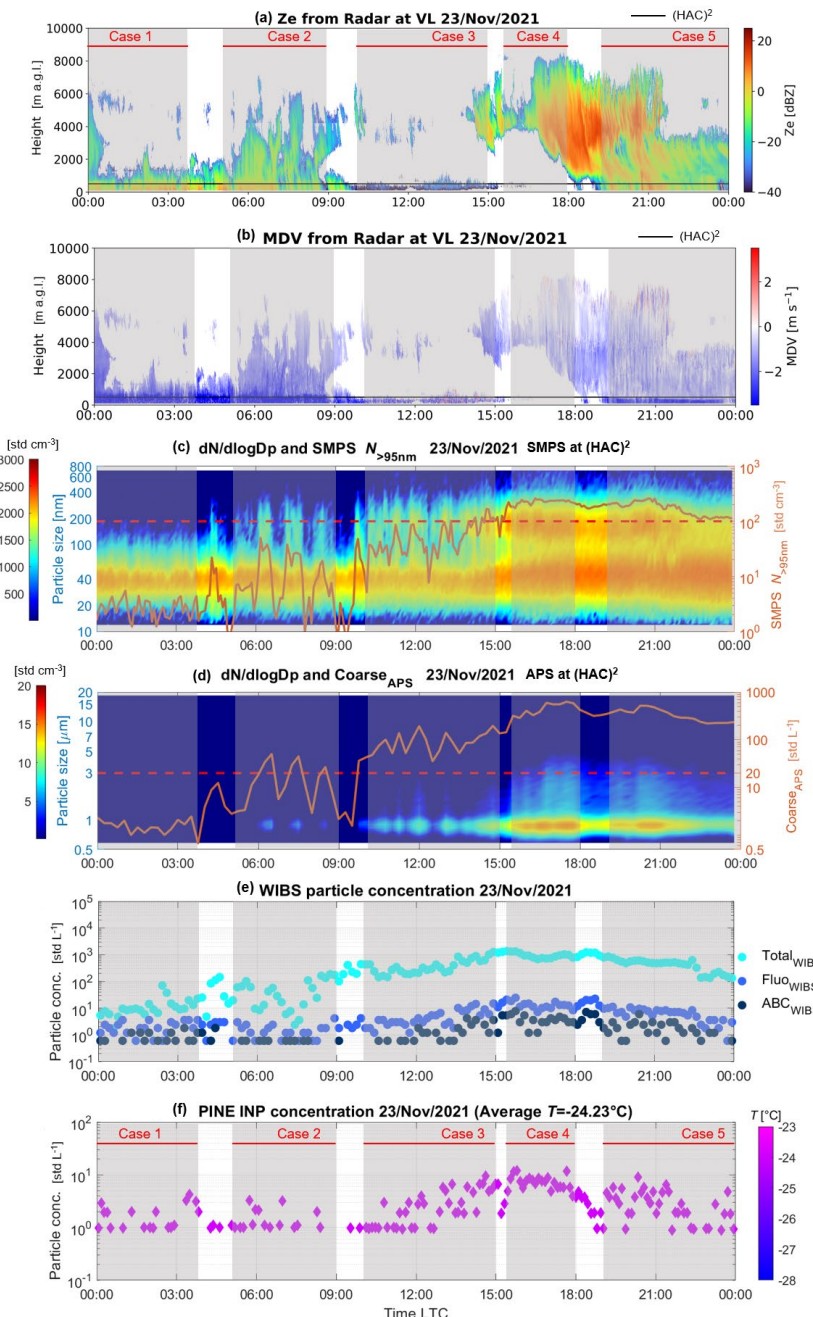

**Figure 11. Timeseries of precipitation condition, aerosol particle and INP concentration on November 23, 2021. (a) and (b) Ze and MDV measured by the radar at VL, respectively. The (HAC)² level is indicated by the black line in the panel. (c) Particle size distribution measured by SMPS at (HAC)² for particles smaller than 800 nm (mobility diameter) and number concentration of SMPS $N_{>95nm}$ particles used to evaluate (HAC)² position with respect to PBL. The left axis shows the size for particle size distribution**



**colour map and the right axis scales SMPS $N_{>95nm}$ values. (d) Particle size distribution measured by APS at (HAC)$^2$ for particles having a size between 0.5 and 20 μm (aerodynamic diameter) and number concentration Coarse$_{APS}$ particles (> 1.0 μm). The left**
**axis shows the size for particle size distribution colour map and the right axis scales Coarse$_{APS}$ values. (e) Total$_{WIBS}$, Fluo$_{WIBS}$ and ABC$_{WIBS}$ particle number concentrations recorded by WIBS at the (HAC)$^2$. (f) PINE INP concentration measured at (HAC)$^2$ with time resolution of ~6 min. The temperature for PINE IN experiments is indicated by the marker colour scaled to the colour bar.**

When (HAC)$^2$ is around PBL and close to the cloud top (case 3), Coarse$_{APS}$ particle concentration increases to a level larger than 20 std L$^{-1}$ (Fig. S12b) because of increased influence from PBL aerosols. The adoption of aerosols from the PBL is
supported by the occasional updrafts shown in Fig. 11b and by the increasing SMPS $N_{>95nm}$ close to 100 std cm$^{-3}$. For the two scenarios of (HAC)$^2$ in PBL with and without precipitation/clouds (case 4 and 5 respectively), Coarse$_{APS}$ particle concentration is well above 20 std L$^{-1}$. The presence of precipitation/clouds leads to a decrease in Coarse$_{APS}$ particle concentration, showing a decreased median value from 414.1 to 330.2 std L$^{-1}$ (Fig. S12b). This suggests the wet removal effects of precipitation on coarse mode particles in the PBL. However, the presence of precipitation/clouds causes an increase in SMPS $N_{<95nm}$ particles
(Figs. S11, 11c and S12c), suggesting that the effect of precipitation/clouds in PBL may also include new particle formation (Khadir et al., 2023).

In general, it can be summarized that Coarse$_{APS}$ particle concentration increases when (HAC)$^2$ is deeper inside the PBL (indicated by a larger SMPS $N_{>95nm}$; Fig. S12b), suggesting that the PBL is the major source for aerosol particles at (HAC)$^2$ on November 23. With increased influences from the PBL, the increases in both ABC$_{WIBS}$ and eBC particles (Fig. S12e and f)
suggest that ABC$_{WIBS}$ particles are relevant for biological particles and eBC emissions are mainly from the PBL. However, the occurrence of precipitation/clouds in the PBL leads to decreased ABC$_{WIBS}$ particles, which is opposite to the effect on aerosols in the FT (see Section 3.1.2). This may result from the wet removal effects of precipitation/clouds on ABC$_{WIBS}$ particles which may dominate over any production of ABC$_{WIBS}$ from precipitation splash (Khadir et al., 2023). The occurrence of precipitation/clouds in the PBL results in a small decrease in eBC mass (Fig. S12f), from 0.07 μg std m$^{-3}$ to 0.05 μg std m$^{-3}$
(median), suggesting a slight wet deposition of eBC particles during the precipitation/clouds periods.

### 3.3.3 INPs observed under different scenarios

INP concentration for (HAC)$^2$ in FT with precipitation (case 1) in Fig. 12a shows a median value of 1.0 std L$^{-1}$, consistent with Fig. 9a for the case of (HAC)$^2$ in FT with precipitation/clouds sampled from the other similar periods during CALISHTO. Compared to case 1 (Fig. 12a), case 2 shows a slightly lower INP concentration (median value 0.6 std L$^{-1}$), attributed to the
decreased availability of Fluo$_{WIBS}$ and ABC$_{WIBS}$ particles (Fig. 11e, S12d and S12e). The results also show that Fluo$_{WIBS}$ and ABC$_{WIBS}$ particles are more important sources of potential INPs than Coarse$_{APS}$, given that a tenfold increase in Coarse$_{APS}$ particles does not lead to an increase in INPs for case 2.

When (HAC)$^2$ is around the PBL and close to cloud top (case 3), it shows an increase in INPs compared to both case 1 and case 2 when (HAC)$^2$ is in the FT. The increased INPs may be attributed to the increased availability of Fluo$_{WIBS}$ and ABC$_{WIBS}$
particles, probably Coarse$_{APS}$ particles as well. When the (HAC)$^2$ is in the PBL (case 4), the INP concentration reaches a peak during the day with a median of 7.2 std L$^{-1}$. A short period of cloudiness around (HAC)$^2$ after 18:00 and the presence of




precipitation/clouds at the $(HAC)^2$ (in the PBL) around 19:00 lead to a decrease in INPs (Fig. 11), likely because of the wet removal effects on aerosols particles. This is different from the results presented in Fig. 9a showing enriched INPs after a period of precipitation/clouds. The different INP changes caused by precipitation in the FT and in the PBL can be attributed

to the corresponding increase and decrease in $ABC_{WIBS}$ particles in the FT (Fig. 9f) and in the PBL (Fig. S12e) respectively, highlighting the importance of $ABC_{WIBS}$ particles in regulating INP abundance under different atmospheric conditions. When $(HAC)^2$ is in the FT (Fig. 9a) where background INPs are rare, $ABC_{WIBS}$ particles produced by precipitation may easily play a pronounced role in influencing INP concentrations. In this case, $ABC_{WIBS}$ particles may come from cloud-processed particles released from hydrometeors in the precipitation/clouds (Prenni et al., 2013), biological particles or soil dust containing

biological components produced by precipitation upon hitting the vegetated or soil surface (Conen et al., 2011; Conen et al., 2017). Differently, precipitation/clouds enriched INPs take a small proportion in the total INPs when $(HAC)^2$ is in the PBL (Fig. 12). Instead, the wet removal effect of precipitation/clouds for $(HAC)^2$ in the PBL may play the major role and remove some $ABC_{WIBS}$ particles in coarse mode that might be active INPs. Therefore, the overall effect of precipitation/clouds on INPs observed at $(HAC)^2$ shows a decrease when $(HAC)^2$ lies within the PBL.

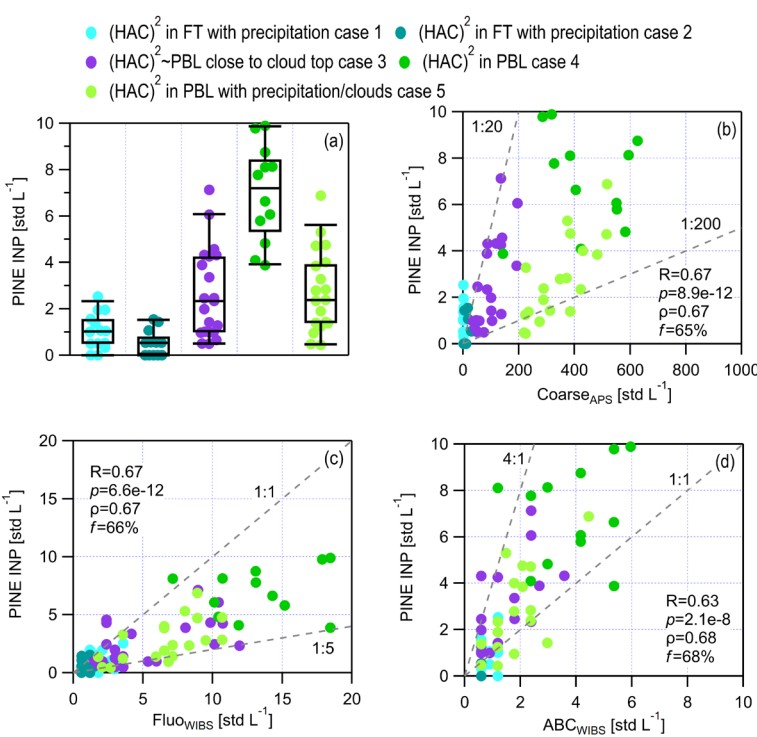


**Figure 12. INP concentration and the relation between INPs and aerosol particles under different scenarios. (a) Box plots of PINE INP abundance of different scenarios classified on November 23. (b) Scatter plots for PINE INP and Coarse$_{APS}$ particle concentration. (c) Scatter plots for PINE INP and Fluo$_{WIBS}$ particle concentration. (d) Scatter plots for PINE INP and ABC$_{WIBS}$ particle concentration. The Pearson correlation coefficient ($R$) and corresponding $p$ value calculated from F–test, and Spearman's**

**rank coefficient ($\rho$), are provided to evaluate the correlation between INP abundance and the particle partitioning. The $P$ value is the probability of obtaining an $R$ value no smaller than the true $R$ value if there is no liner correlation between INPs and the given parameter. The value of $f$ means the percentage of data points within the range confined by dashed lines in the panel.**



Figure 12 also shows that the INP concentration observed at (HAC)[2] on November 23 generally has a significantly positive correlation with $Coarse_{APS}$, $Fluo_{WIBS}$ and $ABC_{WIBS}$ particle concentration. The concentration of $Coarse_{APS}$ particles is generally
higher than that of INPs by > 20 times (> 65% in Fig. 12b). Also, 66% INP concentration values are lower than those of $Fluo_{WIBS}$ (Fig. 12c) whereas 68% INP concentration values are larger than those of $ABC_{WIBS}$ (Fig. 12d). The results mean that INPs are from coarse particles (> 1.0 μm) showing fluorescence whereas $ABC_{WIBS}$ particles are not the only source for INPs.

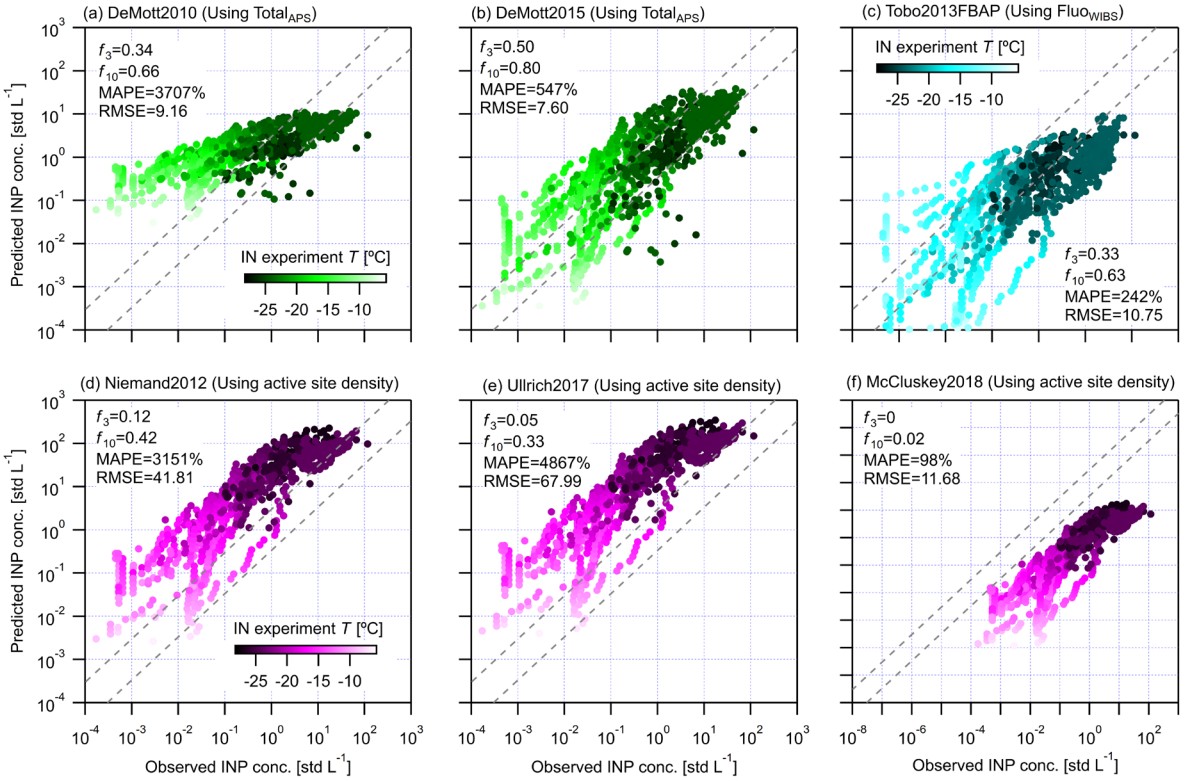

**Figure 13. Scatter plots of observed INP concentration and the concentration calculated by parameterizations (in Table 1) from the**
**literature. Parameterizations in panel (g) to (i) are based on the overlapped WIBS, APS and SMPS dataset. (a) DeMott2010. (b) DeMott2015. (c) Tobo2013FBAP. Here, $Fluo_{WIBS}$ is used to substitute $n_{FABPS}$ measured by UV-APS as used in Tobo et al. (2013). (d) Niemand2012. (e) Ullrich2017. (f) McCluskey2018. The temperature condition for INP data is scaled to the colour bar. Parameterizations using the same aerosol properties use the same colour bar. The dashed lines confine the range for observed and predicted data points within a factor of 3. The fractions of observed and predicted data points within a factor of 3 ($f_3$) and 10 ($f_{10}$)**
**are provided in each panel, respectively. MAPE stands for Mean absolute percentage error. RMSE is the root-mean-square error used as a measure of the difference between observed and predicted data.**

### 3.4 INP parameterization

### 3.4.1 Predicting INPs observed at Mt. Helmos using published parameterizations

A variety of parameterizations, summarized in Table 1, have been proposed to estimate $N_{INP}$ using aerosol properties, such as
particle number concentration and particle surface area. We evaluate their ability to reproduce the observed $N_{INP}$ at Mt. Helmos. Here, we note that $Fluo_{WIBS}$ is used to substitute $n_{FBAPS}$ (the number concentration of fluorescent aerosol particles monitored



by an UV-APS) used in Tobo et al. (2013). The predictability of those parameterizations is evaluated by comparing $N_{INP}$ observations to the calculated $N_{INP}$ results. The evaluation of the predictability of each INP parameterization (in Table 1) for INPs from different INP sources discussed in Section 2.4 is presented in Figs. S14 to S19. In addition, we note that INSEKT INPs are of lower estimations compared to PINE INPs beyond a factor of 5 (see Supplement S3) are excluded from the parameterization dataset.

**Table 3. Different parameterizations proposed for predicting INPs at Mt. Helmos.**

| INP Parameterizations | Used aerosol property | Formulation |
|---|---|---|
| Helmos DeMott2015 | $Total_{APS}$ (std cm$^{-3}$) | $N_{INP} = cf * Total_{APS}^{(-aT+b)} * exp(-cT + d)$<br>($cf = 3$, $a = 0.09$, $b = -1.05$, $c = 0.49$, $d = -12.66$; $T$ in °C) |
| Helmos Total$_{APS}$ | | $N_{INP} = a(-T)^b * (Total_{APS}/1000)^{(-cT+d)} * (APS_{ratio} * (eT + f))$<br>($a = 900$, $b = -9.56$, $c = 0.14562$, $d = -2.769$, $e = -32000000$, $f = 860000$) |
| Helmos Tobo2013FBAP | Fluo$_{WIBS}$ (std cm$^{-3}$) | $N_{INP} = Fluo_{WIBS}^{(-aT+b)} * exp(-cT + d)$<br>($a = 0.096$, $b = -1.49$, $c = 0.96$, $d = -18.9$; $T$ in °C) |
| Helmos Fluo$_{WIBS}$ | | $N_{INP} = exp(aT + b) * (Fluo_{WIBS}/1000)^{(cT+d)} * (WIBS_{ratio})^{(eT+f)}$<br>($a = -0.096725$, $b = -3.6932$, $c = -0.16288$, $d = -3.04$ $e = 0.024358$, $f = 0.44052$; $T$ in °C) |
| Helmos Total$_{WIBS\_1}$ | Total$_{WIBS}$ (std L$^{-1}$) | $N_{INP} = exp(aT + b) * Total_{WIBS}^{(cT+d)} * (WIBS_{ratio} * e + f)$<br>($a = 0.45678$, $b = -3.456$, $c = -0.15$, $d = -2.7989$, $e = 220000$, $f = 18400$; $T$ in °C) |
| Helmos Total$_{WIBS\_2}$ | | $N_{INP} = a(-T)^b * Total_{WIBS}^{(cT+d)} * (\dfrac{1}{WIBS_{ratio} * e + f})$<br>($a = -4244444.44$, $b = -5.5$, $c = -0.119$, $d = -1.69$, $e = 641.55$, $f = -154.31$; $T$ in °C) |
| Helmos Total$_{SMPS+APS\_1}$ | Total$_{SMPS+APS}$ (std cm$^{-3}$) | $N_{INP} = cf * Total_{SMPS+APS}^{(-aT+b)} * exp(-cT + d)$<br>($cf = 1e-5$, $a = 0.04$, $b = -0.19$, $c = 0.2$, $d = 2$; $T$ in °C) |
| Helmos Total$_{SMPS+APS\_2}$ | | $N_{INP} = a(-T)^b * Total_{APS}^{(-cT+d)} * (SMPS\_APS_{ratio} * (eT + f))$<br>($a = 9000000$, $b = -10.3$, $c = 0.16$, $d = -2.769$, $e = -3200000$, $f = 800$; $T$ in °C) |
| Helmos S$_{SMPS+APS}$ | S$_{SMPS+APS}$ (std m$^2$ cm$^{-3}$) | $N_{INP} = 1000 * S_{SMPS+APS} * exp(aT + b)$<br>($a = -0.5$, $b = 6.6$; $T$ in °C) |

Note. APS$_{ratio}$ means the ratio of Coarse$_{APS}$ to Fine$_{APS}$ particle concentration. WIBS$_{ratio}$ means the ratio of Fluo$_{WIBS}$ to NonFluo$_{WIBS}$ particle concentration. SMPS_APS$_{ratio}$ means the ratio of Coarse$_{APS}$ to Fine$_{SMPS+APS}$ particle concentration

DeMott2015 can predict 80% data points within a factor of 10 compared to observations (Fig. 13b), which is better than DeMott2010 (66%, in Fig. 13a). Generally, overestimated INP data points by DeMott2015 are more than those underestimated (Fig. 13b). For Tobo2013FBAP, we first note that Fluo$_{WIBS}$ particle concentration used in this study is larger than the fluorescent particle concentration measured by UV-APS as used in Tobo et al. (2013). Such an input difference would have led to an overprediction in INPs, if data from the Mt. Helmos were similar to those observed in Tobo et al. (2013). We find however, that Tobo2013FBAP generally underestimates INPs at Mt. Helmos, especially for temperatures lower than −20 °C when dust particles dominate the INP sources (Fig. S16h and i). This discrepancy may be explained when considering that Tobo2013FBAP is developed based on an INP population of biological particles that activate as ice at warm temperatures (> −15 °C). Given that the abundance of biological particles is lower than that of dust particles, thus, Tobo2013FBAP



underestimates INPs originated from dust particles activating ice at lower temperatures. Nevertheless, Tobo2013FBAP is able

to agree with 63% of INP observations in Mt. Helmos within two orders of magnitude (Fig. 13c).

The results in Fig. 13d to 13f are based on the subset of data when both SMPS and APS data (hereafter $INP_{SMPS+APS}$ data) are available. Both Niemand2012 and Ullrich2017 systematically overestimate $INP_{SMPS+APS}$ (Fig. 13d and 13e), which may be attributed to the data base used for the development of both parameterizations (dust samples tested in laboratory studies), which may have exhibited a larger active site density compared to the atmospheric particles investigated in this study. On the

contrary, McCluskey2018 systematically underestimates the observed $INP_{SMPS+APS}$ data (Fig. 13f), likely because it is based on sea spray aerosol, which may have a lower active site density compared to INPs observed in this study.

Of all literature parameterizations tested, DeMott2015 is the best to predict INPs at Mt. Helmos. Its comparatively good performance can be attributed to its larger and more inclusive data base from both laboratory and field experiments. DeMott2015 will therefore be adapted in Section 3.4.2 to optimize its applicability for Mt. Helmos by proposing new

parameters.

### 3.4.2 Parameterizations for INPs using the CALISHTO data

Nine parameterizations using different aerosol properties are developed (Table 3), including parameterizations adapted from the literature and proposed ones depending on the observed relations between INPs and aerosol properties. We first adapt the DeMott2015, named "Helmos DeMott2015", with a new set of parameters calculated by fitting the formula to the relevant

data collected at Mt. Helmos. Considering that the IN ability of aerosol sources show a significant correlation with $Coarse_{APS}$ to $Fine_{APS}$ ratio (Fig. 10c), thus the ratio, termed $APS_{ratio}$ hereafter, is included in a new INP parameterization termed "Helmos $Total_{APS}$". We also adapt Tobo2013FBAP with new parameters and it will be compared to a new parameterization we develop, termed "Helmos $Fluo_{WIBS}$" that predicts INP as a function of $Fluo_{WIBS}$, $WIBS_{ratio}$ ($Fluo_{WIBS}$ to $NonFluo_{WIBS}$) and $T$. Given that $Fluo_{WIBS}$ may not include all potential INPs (especially for $T < -20$ °C where nonbiological particles dominate), we propose

two parameterizations ("Helmos $Total_{WIBS}\_1$" and "Helmos $Total_{WIBS}\_2$") using $Total_{WIBS}$ to represent aerosol particles that may serve sources for INPs. Both parameterizations depend on $Total_{WIBS}$, $WIBS_{ratio}$ and $T$ but with different formula forms. Moreover, $Total_{SMPS+APS}$ is used as the input for INP source particles to calculate $N_{INP}$ to include potentials in a larger size range, given that particles smaller than 0.5 μm may also be relevant for INPs (Kanji et al., 2017). With and without including the ratio of $Coarse_{APS}$ to $Fine_{SMPS+APS}$ particles ($SMPS\_APS_{Ratio}$), two parameterizations ("Helmos $Total_{SMPS+APS}\_1$" and

"Helmos $Total_{SMPS+APS}\_2$") are proposed to calculate $N_{INP}$ based on $Total_{SMPS+APS}$ and $T$. We also provide parameters for a surface-area based parameterization ("Helmos $S_{SMPS+APS}$") using the concept of surface-active sites. In the following, we discuss the performance of INP parameterizations introduced above (Fig. 14) and evaluate the predictability of each INP parameterization (Figs. S20 to S35) for INPs from different INP sources discussed in Section 2.4.

*INP parameterizations using $Total_{APS}$ particle concentration.* The results in Fig. 14a and b compare the predictability of

Helmos DeMott2015 and Helmos $Total_{APS}$. After adaption, the percentage of $N_{INP}$ data points within two orders of magnitude





compared to observations increases by 16% for Helmos DeMott2015 (96% in Fig. 14a) in comparison to DeMott2015 (80%

**Figure 14. Different parameterizations for predicting INPs at (HAC)².** Parameterizations in panel (a) to (f) are based on the dataset when both APS and WIBS are available. Parameterizations in panel (g) to (i) are based on the overlapped APS and WIBS dataset.
(a) Helmos DeMott2015. (b) Helmos Total$_{APS}$. (c) Helmos Tobo2013FBAP. (d) Helmos Fluo$_{WIBS}$. (e) Helmos Total$_{WIBS}$_1. (f) Helmos Total$_{WIBS}$_2. (g) Helmos Total$_{SMPS+APS}$_1. (h) Helmos Total$_{SMPS+APS}$_2. (i) Helmos S$_{SMPS+APS}$. The temperature condition for INP data is scaled to the colour bar. Note that parameterizations using the same aerosol properties use the same colour bar. The dashed lines confine the range for observed and predicted data points within a factor of 3. The fraction of observed and predicted data points within a factor of 3 ($f_3$) and 10 ($f_{10}$) is provided in each panel, respectively. MAPE stands for mean absolute percentage error. RMSE is the root-mean-square error used as a measure of the difference between observed and predicted data. BIC is a value calculated by applying the Bayesian information criteria to evaluate the goodness of parameterizations based on the same dataset (Schwarz, 1978).



in Fig. 13a). Also, ~11% of the 16% comes from the predictions on $N_{INP}$ values within a factor of 3 compared to the observation. However, Helmos DeMott2015 does not show improvements in predicting INPs for aerosol sources in the FT, as indicated by
the underestimated values in the red square in Fig. 14a (also Fig. S20b and c). In particular, as shown in Fig. S20c, only 41% predictions from Helmos DeMott2015 are within a factor of 10 compared to the observations when $(HAC)^2$ is in the FT with influences of precipitation/clouds. Figure 14b shows that the consideration of $APS_{ratio}$ into Helmos $Total_{APS}$ parameterization can improve its predictability by increasing the $f_3$ from 0.61 to 0.70, compared to Helmos DeMott2015. Also, the lower mean absolute percentage error (MAPE), root-mean-square error (RMSE) and Bayesian information criteria (BIC) values of Helmos
$Total_{APS}$ suggest its slightly better performance than Helmos DeMott2015, by showing a smaller deviation, a smaller overall error and a larger formula fitting goodness, respectively. However, Helmos $Total_{APS}$ cannot predict INPs within the $f_{10}$ range at warm temperatures (> −20 °C) probably for biological particles from continental aerosols (Garcia et al., 2012; Pummer et al., 2015), as marked by the red square in Fig. 14b. Those data points are from two INSEKT filter samples collected on October 28 and 29 when $(HAC)^2$ is affected by continental aerosols (Fig. S21i).

*INP parameterizations using $Fluo_{WIBS}$ particle concentration.* With a new set of parameters, Helmos Tobo2013FABP in Fig. 14c shows improved predictability for $N_{INP}$, showing increased $f_3$ and $f_{10}$ by 0.24 and 0.33 respectively, in comparison to Tobo2013FABP in Fig. 13c. This suggests INP populations observed at Mt. Helmos might have different IN abilities from those observed in Tobo et al. (2013). After incorporating $WIBS_{ratio}$ into Helmos $Fluo_{WIBS}$, the parameterization can further increase the $f_3$ predictability to 0.70 (Fig. 14d) compared to Helmos Tobo2013FABP. Also, Helmos $Fluo_{WIBS}$ shows lower
RMSE and BIC values, suggesting a slightly better fitting goodness. On the other hand, Helmos $Fluo_{WIBS}$ shows a larger MAPE value, which is due to its more scattered predictions on INPs at warm temperatures (> −20 °C).

*INP parameterizations using $Total_{WIBS}$ particle concentration.* Helmos $Total_{WIBS}\_1$ and $Total_{WIBS}\_2$ show comparable $f_3$ and $f_{10}$ predictions larger than 0.67 and 0.95 respectively (Fig. 14e and f). Compared to Helmos $Total_{WIBS}\_1$, Helmos $Total_{WIBS}\_2$ shows smaller values of MAPE and RMSE, suggesting an overall slightly smaller deviation of $N_{INP}$ predictions. Except for
INPs from the source of South dust in PBL after marine aerosols, Helmos $Total_{WIBS}\_1$ can predict $N_{INP}$ from the other sources with a $f_{10}$ larger than 97% (Fig. S24). The exceptional performance for the source of South dust in PBL after marine aerosols may be due to the limited observations, for which other parameterizations show similar results (see results in Supplement S9). Compared to Helmos $Total_{WIBS}\_1$, the slightly overall lower $f_3$ and $f_{10}$ predictability of Helmos $Total_{WIBS}\_2$ is because of its relatively poor predictions for INPs from the source of aerosols in the FT with precipitation/clouds (Fig. S25c).

*INP parameterizations using $Total_{SMPS+APS}$ particle concentration.* Figure 14g shows a $f_3$ and $f_{10}$ prediction of 0.54 and 0.83 respectively for Helmos $Total_{SMPS+APS}\_1$. Its $N_{INP}$ prediction skill is the lowest amongst parameterizations shown in Fig. 14, possibly due to its poor ability to predict dust INPs (Fig. S26h). Compared to Helmos $Total_{SMPS+APS}\_1$, the inclusion of $SMPS\_APS_{Ratio}$ into Helmos $Total_{SMPS+APS}\_2$ helps the parameterization to capture INPs from different particle populations and increases the $f_{10}$ prediction by 7% (Fig. 14h). Predictions from Helmos $Total_{SMPS+APS}\_2$ also show a more symmetric
distribution around the 1:1 line. However, Helmos $Total_{SMPS+APS}\_2$ shows a decreased predictability for INPs from the source





*INP parameterizations using $S_{SMPS+APS}$ total particle surface area.* The parameterization, based on active site density (Young, 1974; Demott, 1994; Connolly et al., 2009), is expressed as a function of $T$ and the particle surface area ($S_{SMPS+APS}$). Compared

to Niemand2012, Ullrich2017 and McCluskey2018 (Table 1), the parameter b for Helmos $S_{SMPS+APS}$ (Table 3) is adjusted to obtain a better prediction, which enables Helmos $S_{SMPS+APS}$ to predict more than 90% data points within a factor of 10 compared to observations (Fig. 14). Helmos $S_{SMPS+APS}$ can also predict INPs from different aerosol sources with a $f_{10}$ larger than 93% (Fig. S28), except INPs sourced from South dust in PBL after marine aerosols.

*Summary of parameterizations and recommendations.* All parameterizations presented in Fig. 14 can predict >80% of the

measured INP concentrations at Mt. Helmos within a factor of 10 of all observations of the corresponding dataset (see the caption). Comparing all parameterizations with the same sub-dataset in Fig. 14, four newly proposed parameterizations, including Helmos Total$_{APS}$, Helmos Fluo$_{WIBS}$, Helmos Total$_{WIBS}$_1 and Helmos Total$_{WIBS}$_2, show $f_3$ and $f_{10}$ predictions approximately 70% and 95% respectively. The superior performance of these four parameterizations is likely because the incorporation of aerosol particle ratios which implicitly are linked to specific particle types with distinct INP characteristics,

hence leading to better predictions of IN activity of particles. Helmos Total$_{WIBS}$_2 has the smallest BIC value, suggesting it may have the highest overall optimality in terms of the model dimension and the maximum likelihood of the prediction (Schwarz, 1978). Notably, only Helmos Total$_{WIBS}$_1 can predict INPs from all different sources with a $f_{10}$ prediction larger than 95%, which suggests its broad applicability for INPs from various sources. In addition, we further evaluate and compare parameterizations in Fig. 14a to f by using them to predict a larger dataset than the sub-dataset used for their developments

(Figs. S29 to S35 in supplement S12). The results suggest that Helmos Total$_{WIBS}$_1 outperforms the others by showing a $f_{10}$ INP prediction larger than 95% for all aerosol sources (except INPs sourced from South dust in PBL after marine aerosols) also for the larger testing dataset (Fig. S34), and hence, it is the parameterization of choice for models that can constrain all the required input. Without including aerosol fluorescent properties, the results in both Fig. S30d (for Helmos DeMott2015) and Fig. S31d (Helmos Total$_{APS}$) suggest that INP parameterizations considering only APS data cannot predict INPs from

continental aerosols at warm temperatures, which are likely of biological origin. If considering only the aerosol particle number concentration for INP prediction, Helmos Tobo2013FBAP outstands Helmos DeMott2015 and Helmos Total$_{SMPS+APS}$_1 (Fig. S22) in predicting more than 90% INPs from different sources, except showing a $f_{10}$ of 0.5 for the dust case.

## 4 Summary and conclusions

Here we study the characteristics of different aerosol sources and their INP abundances in mixed-phased clouds at the high

altitude Helmos Hellenic Atmospheric Aerosol and Climate Change station ((HAC)[2]) at Mount Helmos, Greece, during the Cloud-AerosoL InteractionS in the Helmos background TropOsphere (CALISTHO) campaign from a period between October and November 2021. INPs were measured for $T > -27$ °C with online (PINE, in all IN mechanisms) and offline (INSEKT, in





the immersion freezing mode) techniques. The relationship between INPs and different aerosol particle types was unravelled to understand the ice formation ability of particles from different aerosol sources. Specific INP source apportionment was carried out using a synergy of in-situ aerosol property characterization measurements, remote sensing and modelling experiments. The relative position of (HAC)$^2$ in the atmosphere with respect to planetary boundary layer (PBL) was determined by using both PBL height results from a pulsed Doppler scanning lidar system (HALO) and the number concentration of particles larger than 95 nm measured by a scanning mobility particle sizer (SMPS $N_{>95nm}$), so as to differentiate the airmass sampled from the PBL to free troposphere (FT). The presence of precipitation/clouds at (HAC)$^2$ was also monitored by a Wide-band Doppler spectral zenith profiler (WProf) radar by using equivalent radar reflectivity factor (Ze) and mean Doppler velocity (MDV). The size distribution (SMPS and APS, aerodynamic particle sizer), optical properties (nephelometer) and chemical composition (ToF-ACSM, time-of-flight aerosol chemical speciation monitor) of different aerosol particle populations were characterized when analysing the footprints and back trajectories of aerosol particles using FLEXPART and HYSPLIT results. Such a comprehensive synthesis study on INP source apportionment is not often reported in the literature. We conclude the following key findings:

- Specific and detailed INP source apportionment was achieved, demonstrated by the distinct characteristics of different sources, including continental, marine and dust aerosols, as well as a mixture of continental and dust aerosols. Fine particles (<1.0 μm) dominate continental aerosols and lead to an Ångström exponent (α) larger than 2.0 whereas coarse particles (>1.0 μm) take a substantial proportion in dust aerosols showing a fairly constant α of approximately 1.0. Marine aerosols show the highest content of Cl$^-$. The distinction of different aerosols is also indicated by their different particle portioning conditions, such as fine-to-coarse particle ratio and fluorescent-to-nonfluorescent particle ratio.

- In addition to PBLH and SMPS $N_{>95nm}$ used to diagnose the PBL condition of the observation site, we note that a threshold value of 20 std L$^{-1}$ for Coarse$_{APS}$ particle concentration can be used as a more conservative standard to qualitatively examine the observation site condition with respect to the PBL. A higher (lower, compared to 20 std L$^-$) value than the threshold indicates the observation site is inside (outside) the PBL. Under strictly defined FT conditions without influences of local and remote aerosols, i.e., background condition, only 1 in 10$^6$ of all aerosol particles acts as INPs for $T = -25 \pm 1.0$ °C.

- Notably, the particle property and INP abundance of different aerosol sources also depend on the atmospheric condition, i.e., the relative position with respect to the PBL. Compared to the condition in the PBL, continental aerosols in the FT contain less particles. In particular, the decrease in coarse particles (>1.0 μm) for continental aerosols in the FT is more significant than that decrease in fine particles (<1.0 μm), which is responsible for its reduced INP abundance compared to the condition in the PBL. The INP abundance in North continental aerosols is more than one order of magnitude higher than that of background condition in the FT when the air mass is in the PBL. Marine aerosols above PBL show a similar INP abundance to that of North continental aerosols above PBL. Dust



containing aerosols present the highest INP abundance among all sources. When dust is mixed with continental aerosol, the INP abundance decreases because of dilution, deposition and deactivation by atmospheric aging.

- The effects of precipitation/clouds lead to a slightly decrease in coarse-sized aerosol particles when the observation site is in the FT, however, it results in a small increase in the INP abundance. Such an enrichment in INPs is attributed to the release of cloud-processed particles and/or near ground particle production caused by precipitation splash, both of which supply active INPs. On the other hand, the presence of precipitation/clouds results in decreased INP abundance when the observation site is in the PBL because of the wet removal of aerosol particles.

- The INP attribution results suggest that biological particles from the PBL air masses may be important INP sources for continental and marine aerosols whereas coarse-sized dust particles play the primary role in the observed INPs in dust aerosols. The concentrations of INPs from marine and continental aerosols above the PBL are within a range of a factor of 5 compared to $ABC_{WIBS}$ particles for more than 93% of the PINE observed INPs, suggesting that biological particles likely make an important contribution to the activated INPs. For dust-containing sources, biological particles may play a secondary role in the observed INPs after dust particles. We highlight that fluorescent particle concentration recorded by WIBS (including biological particles and non-biological particles showing fluorescence) is able to describe all PINE observed INP concentration (for $T = -24 - -27$ °C) within a factor of 5 for more than 80% observed data points. Additionally, we note that mineral dust and carbonaceous particles may fluoresce because of some association with biological material or PAHs from combustion. Both eBC and dust particles can be transported by a long distance in the FT.

- The IN ability of different aerosol sources shows distinguishable characteristics and presents correlations with $Coarse_{APS}$ and fluorescent particle proportions. The above new findings were used to improve the predictability of INP parameterizations, which predict observed INPs for more than 90% data points for all identified INP sources and outperform existing INP parameterizations in the literature. The APS based INP parameterizations (Helmos $Total_{APS}$ in Table 3) show a less satisfactory INP predictability for warm temperatures ($> -20$ °C) compared to their predictability for cold temperatures ($< -20$ °C), suggesting their inadequate capability of capturing biological INPs. Using particle fluorescence characteristics can improve INP parameterization predictability at warm temperatures. The parameterization developed from a sub-dataset (~80% of observations) was evaluated by predicting the whole observations with both WIBS and INP data. Considering biological particles that fluoresce and the proportion of those particles, our INP parameterization (Helmos $Total_{WIBS}\_1$ in Table 3) is able to predict the INP abundance in different sources within a factor of 10 for more than 95% data points, thus outperforms the existing parametrizations in the literature and the others proposed in this study.

- Existing INP parameterizations in the literature may be improved by including the ratio of fluorescent-to-nonfluorescent or coarse-to-fine particles if available. In doing so, the predictability of published parameterizations and their applicability for climate models can be improved. Thus, MPC simulations for regions with different INP sources, such as a cross-road of different air masses like (HAC)$^2$, or for different regions with different prevailing



INP sources, such as continental or marine regions, can be better achieved based on published dataset and parameterizations from existing field studies. Ultimately, the regional and global simulations of aerosol-cloud interactions, as well as climate modelling, can be benefited from the improved INP parameterization by using its own dataset.

The underestimation of INPs from INSEKT experiments, because of insufficient aerosol particle collection during aerosol
sampling and/or filter washing, may partly influence the INP parameterizations. In this regard, INSEKT data, that are analysed as lower estimations by more than a factor of 5 compared to PINE INPs (Section 3.4.1 and Supplement S3), was filtered before using for parameterization. For future studies, we suggest more flexible aerosol sampling strategies for offline INP measurements other than that used in this study (~1 day for INSEKT), like by employing automated filter samplers that allow sampling periods adjustable according to the changing aerosol particle concentrations, or by using wet cyclones to collect
aerosol particles in liquids. The optimized sampling strategies for offline INP measurements should avoid filter over loading during dust events. To further test and validate the predictability of WIBS data based INP parametrizations, it necessitates a larger dataset that spans different seasons and based on observations at different sites in different regions. The further development of WIBS data based INP parameterizations will be addressed in our following work.

*Author contributions.* AN, AP and KE organized the CALISHTO campaign. KG and AN conceived and led this study. KG led the analysis, wrote the original manuscript together with AN and prepared all the figures with contributions from RF, AB and SV. KG analysed the data and interpreted the results with input from AN, AP, RF and PG. AN, AP, KE, RF, AB, FV, SV, MIG, OZ and PF conducted experiments and collected the raw data. SV performed FLEXPART simulations. KG ran HYSPLIT with support from AP. All authors discussed, reviewed and edited the manuscript.


*Financial support.* This research has been supported by the European Research Council PyroTRACH project (project ID 726165) funded from H2020-EU.1.1. (ERC), the Swiss National Science Foundation project 192292, Atmospheric Acidity Interactions with Dust and its Impacts (AAIDI), the "PANhellnfrastructure for Atmospheric Composition and climatE change" (MIS 5021516). AP and RF acknowledge funding by the Basic Research Program PEVE (NTUA) under contract
PEVE0011/2021.

*Acknowledgements.* We thank Dr. Tobias Könemann from Envicontrol Group for providing us access to a WIBS. We are also grateful to Dr. Ghislain Motos for his help with IGOR toolkit for WIBS data processing. AN thanks Hector Angelos Nenes whom assisted with the installation of the WIBS at Mt. Helmos.


*Competing interests.* The authors declare that they have no conflict of interest.



*Data availability.* The data presented in this publication will be made available at https://www.envidat.ch. The DOI link will be activated for public access upon acceptance of publication.

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
