# Peer review of "Biological and dust aerosol as sources of ice nucleating particles in the Eastern Mediterranean: source apportionment, atmospheric processing and parameterization"

_EGUsphere, 2024_

## Author Comment (AC1)

**Response to EGUSPHERE-2024-511 reviews for RC1**

We thank Reviewer #1 for their effort and feedback on our manuscript EGUSPHERE-2024-511. In response to the suggestions and questions, please find our answers and corrections listed below: **Reviewer #1 comments are extracted in bold from original review supplement**; our responses are given directly below in normal font; *the original text in previous manuscript is repeated in red italic* and *revised text is typed in blue italic*.

**General comments:**

**The manuscript is pleasant to read, also Figures and Tables are clear.**

**There is little I would recommend to change in this manuscript. My main concern are general statements about INPs in which their activation temperature is not mentioned. In this study, INPs include those measured by PINE (ca. -25 °C) and others measured by INSEKT (-5 °C to -25 °C). Throughout the text, it should always be clear which activation temperature applies in a statement. A first example is in the Abstract, line 25: "...approximately 1 in 10^6 aerosol particles serve as INPs." A much later example is on page 30, lines 753 and 754: "Therefore, the overall effect of precipitation/clouds on INPs observed at (HAC)2 shows a decrease when (HAC)2 lies within the PBL." It should be made clear that this finding relates to INPs active at around -25 °C. Testa et al. (2021; https://doi.org/10.1029/2021JD035186) made a similar observation for INPs active at around -25 °C, but at the same time INPs active at -12 °C were found to have increased (see Figure 5 in Testa et al., 2021). Hence, activation temperature matters not only in terms of the number concentration, but also in terms of atmospheric behaviour.**

We thank Reviewer #1 for their positive comments, and thoughtful comments. We also apologize for not clarifying more the INPs temperature condition. The corresponding statements in the manuscript are now revised as below:

a. The original statement in lines 24–25 was as below:

*"When the observation site is in the Free Troposphere (FT), approximately 1 in $10^6$ aerosol particles serve as INPs."*

Now, it is revised as:

*"When the observation site is in the* *free troposphere (FT), approximately 1 in $10^6$ aerosol particles serve as INPs around $-25°C$."*

b. The original statement in lines 754–755 was as below:

*"Therefore, the overall effect of precipitation/clouds on INPs observed at $(HAC)^2$ shows a decrease when $(HAC)^2$ lies within the PBL."*

Now, it is revised:

*"Therefore, the overall effect of precipitation/clouds on INPs observed at $(HAC)^2$ for temperatures around $-24.2°C$ (Fig. 11f) shows a decrease when $(HAC)^2$ lies within the PBL."*

Our results are also in agreement with Testa et al. (2021) and present that the INP abundance observed at $(HAC)^2$ not only depends on the temperature condition for INP observations but also is regulated by the atmospheric condition of the site, the relevant aerosol source properties and meteorological conditions. Those points were individually presented in the conclusion part of the manuscript. We now add a summary sentence in the conclusion part and refer to Testa et al. (2021):

*"Additionally, we note that the INP abundance is also regulated by the property of relevant aerosol source and meteorological conditions, e.g., precipitation (Testa et al., 2021), in addition to temperatures and FT/PBL scenarios."*

**Minor issues**

1. **Page 5, section 2.2.1: Please add to the description of the offline INP observations the filter material, diameter, pore size, and the flow rate of the sampler.**

Good point. The revised statement is:

*"To prepare the freezing aliquots, aerosol particles were first sampled onto filters (0.2 μm Whatman Nuclepore track-etched polycarbonate membranes, 47 mm, with a flowrate of 9 liter per minute) from an omnidirectional total inlet ..."*

**2. Line 214: "40000 air parcels" probably should be "40000 particles"**

The original text is correct. No change made.

**3. Line 245: "take up a large fraction" or "make up a large fraction"?**

Done. Change made ("*make up ...*").

**4. Lines 253 and 254: "to differentiate the difference" I do not understand the meaning of this expression.**

Thank you for making this point. The statement now reads: "*For example, the synergy of in-situ and remote sensing results enables to differentiate the distinct characters of the sources in Fig. 2b and c.*"

**5. Figure 5: Please add x-axes to the plots, even if they only state the running number of observations in each type of aerosol category.**

Good point! Fig.5 is now modified, and the caption is now revised to describe the axes.

**6. Figure 8: I wonder why the number of INPs measured with PINE does not increase with decreasing temperature. Please add a note on this issue to the Figure legend.**

Good point. An extra statement is now added to the legend of Figure 8 as: "*Higher PINE INP concentrations at −24°C compared to lower temperatures are because PINE was run during the Saharan dust event (i.e., the source of South dust in PBL as defined in Fig. 3) whereas INPs at lower temperatures originate from other aerosol sources with lower IN abilities.*"

**7. Figure 9a: The temperature indicated for measurements of "South dust in PBL after marine aerosols" is -2.39. I guess it should be -23.9.**

Done.

**8. Figure 9c and 9d: The effect of precipitation/clouds on INPs in FT is very similar in direction and magnitude as observed in winter in the Swiss Alps by Mignani et al. (2021; https://doi.org/10.5194/acp-21-657-2021).**

Good point! The following statement is now added to the text to reflect this point: "*Also, the results in Fig. 9a, c and d are consistent with Mignani et al. (2021) who reported decreased $Total_{APS}$ and $Coarse_{APS}$ particles but slightly increased INPs after precipitations.*"

**9. Table 2: Readability of p-values would be improved by replacing the scientific notation of very small values (e.g. 9.3e-122) by "< 0.001".**

Thank you for this suggestion! Change made.

**10. Line 548 to 550: A further explanation of why values reported by Lacher et al. (2021) for Jungfraujoch (3580 m) were smaller than what was found at Mt. Helmos (2314 m) could be the higher elevation of Jungfraujoch.**

Good point. A sentence is now added to the text as follows: "*Note that the lower INP concentrations observed at Jungfraujoch compared to (HAC)$^2$ can be attributed to the much higher altitude of the former.*"

**11. Lines 588 and 589: Also in the Arctic, fluorescent particles constitute the vast majority of INPs (active at -15 °C), as Freitas et al. have recently reported (2023; https://doi.org/10.1038/s41467-023-41696-7).**

Good point. A sentence is now added to the text as follows: "*..., and it is consistent with Pereira Freitas et al. (2023) who found fluorescent biological aerosol particles as dominant sources for INPs activating at T around −15°C.*"

**12. Line 725: "...ABCWIBS particles are relevant for biological particles..." I am not sure what is meant by this expression. Do you mean something like "...ABCWIBS particles are related to biological particles..."**

Indeed so! The expression now reads: "*With increased influences from the PBL, the increases in both ABC$_{WIBS}$ and eBC particles (Fig. S12e and f) suggest that ABC$_{WIBS}$ particles are related to biological particles and eBC emissions are mainly from the PBL.*"

**References:**

Mignani, C., Wieder, J., Sprenger, M. A., Kanji, Z. A., Henneberger, J., Alewell, C., and Conen, F.: Towards parameterising atmospheric concentrations of ice-nucleating particles active at moderate supercooling, Atmos. Chem. Phys., 21, 657-664, https://doi.org/10.5194/acp-21-657-2021, 2021.

Pereira Freitas, G., Adachi, K., Conen, F., Heslin-Rees, D., Krejci, R., Tobo, Y., Yttri, K. E., and Zieger, P.: Regionally sourced bioaerosols drive high-temperature ice nucleating particles in the Arctic, Nat. Commun., 14, 5997, https://doi.org/10.1038/s41467-023-41696-7, 2023.

Testa, B., Hill, T. C. J., Marsden, N. A., Barry, K. R., Hume, C. C., Bian, Q., Uetake, J., Hare, H., Perkins, R. J., Möhler, O., Kreidenweis, S. M., and DeMott, P. J.: Ice Nucleating Particle Connections to Regional Argentinian Land Surface Emissions and Weather During the Cloud, Aerosol, and Complex Terrain Interactions Experiment, J. Geophys. Res. Atmos., 126, https://doi.org/10.1029/2021jd035186, 2021.

---

## Author Comment (AC2)

**Response to EGUSPHERE-2024-511 reviews for RC2**

We thank Reviewer #2 for their effort and feedback on our manuscript EGUSPHERE-2024-511. In response to the suggestions and questions, please find our answers and corrections listed below: **Reviewer #2 comments are extracted in bold from original review supplement**; our responses are given directly below in normal font; *the original text in previous manuscript is repeated in red italic* and *revised text is typed in blue italic*.

**General comments:**

**This study provides a comprehensive analysis of the sources of Ice Nucleating Particles (INPs) for mixed-phase clouds at Mount Helmos in the Eastern Mediterranean. By integrating in-situ observations, remote sensing data, and modeling experiments, the study examines the influence of boundary layer turbulence, vertical aerosol distributions, and meteorological conditions. It distinguishes among contrasting meteorological situations where aerosol properties and associated INP activities and levels differ. Additionally, the study evaluates existing INP schemes in the literature and proposes new INP parameterizations, constrained by the observations, that outperform previous ones for this specific location.**

**This is an excellently written and highly relevant study. The extensive experimental data is thoroughly analyzed, and many of the findings are novel. The association between different aerosol properties and INPs is robust, and the inclusion of such dependencies in INP schemes is promising.**

**I have no major concerns about this paper. One possible consideration is its length—39 pages with a supplementary section containing 35 figures. While the paper is comprehensive and detailed, it might have been more digestible as a two-part paper: one focusing on the thorough analyses and the other on the development and evaluation of the INP schemes.**

We appreciate these positive comments! We did consider separating this manuscript into two parts, but preferred the current organization, to have everything in one place.

**Specific comments**

1. **The underestimation of INP concentrations by the INSEKT compared to the PINE is a weakness of the methodology. There is a specific analysis of the differences between the INSEKT and the PINE in the supplemental material and a final discussion in the conclusions. However, it is important to discuss more broadly the implications of such potential systematic error in the evaluation of the INP schemes and the development of new schemes. Providing an estimate of the effect of this underestimation on the evaluation of INP schemes would be helpful. To what extent is this affecting the evaluation of schemes in the literature? How were INPs measured in other studies? This is particularly important when comparing with other studies and for future evaluation of the new schemes proposed with INP data obtained with other methods, as well as when using the new schemes in models.**

We agree. To address the above concerns, we have added the following sentences to the revised Summary and Conclusion part: "*We note that underestimated INPs caused by insufficient sampling of aerosol particles for offline INP analysis may also result in lower estimation of aerosol-cloud interactions for warm temperatures in the MPC regime (e.g., >−15°C) which could carry important implications for inducing biases in the microphysical evolution of clouds in models. Therefore, the particle collection efficiency should be evaluated for existing INP data and INP parameterizations in the literature before using for climate models. Filter sampling (Conen et al., 2015; Schneider et al., 2021) and liquid impinger (Wieder et al., 2022) are commonly used in existing studies to collect INP samples for offline analysis. Standard sampling protocols for both methods are required to ensure sampling efficiency inter-comparison.*"

2. **Figure 5: Understanding the differences on the proportions of the WIBS channels between the aerosol sources from this Figure is challenging. I suggest that you use box plots as in Figure 3. This would provide a better quantitative comparison of the proportions among sources.**

Thanks for the suggestion. It is not a single type of fluorescent particle that makes an identified aerosol source distinct from the others. Instead, the distinctions of fluorescent properties of an aerosol source are shown by the portioning of different types of fluorescent particles. Therefore, we present the fraction pattern of different types of WIBS fluorescent

particles in Fig. 5 but do not directly show the exact values of the properties like in Fig.3. Nevertheless, we now provide box plots (Fig. R1) for different types of WIBS fluorescent particles for all identified aerosol sources (shown below). For example, three aerosol sources influenced by Saharan dust show comparable amount of $B_{WIBS}$ particles (Fig. R1b), however, the median number concentration of $B_{WIBS}$ particles of those three sources varies by approximately one order of magnitude (Fig. R1c). Thus, the fraction patterns presented in Fig. 5 in the original manuscript is more informative than box plots in Fig. R1.

[Figure]

**Figure R1.** Box plots for different types of WIBS fluorescent particles for all identified aerosol sources. (a) $A_{WIBS}$. (b) $B_{WIBS}$. (c) $C_{WIBS}$. (d) $AB_{WIBS}$. (e) $AC_{WIBS}$. (f) $BC_{WIBS}$. The box shows the median line and the range between 25th and 75th quartiles. The lower and upper caps of the box indicate the 9th and 91th percentiles, respectively. Note that there is no WIBS data available for the source of North continental aerosols in PBL.

3. **Section 3.4.1 and Figure 13. I do not fully understand the rationale of comparing existing INP schemes beyond their species-range of applicability. For instance, Niemand and Ulrich focus on dust, McCluskey on marine organics and Tobo on biological particles mostly. While in the supplemental material additional figures are provided for each of the sources and INP schemes, I believe that Figure 13 in the main paper would be more informative if the color coding would refer to the INP source in the observations. Also, please discuss in this section how the uncertainty related to the use of the INSEKT may affect this evaluation.**

This is a good question. We use existing INP parameterizations (Niemand et al., 2012; Tobo et al., 2013; Ullrich et al., 2017; Mccluskey et al., 2018) to test their capability to predict INPs observed at $(HAC)^2$ and understand prediction biases for the different particle types seen at $(HAC)^2$. This is useful to examine the generality and limitations of each parameterization – the insights of which were then used to develop the new formulation presented in our manuscript.

We considered using different colors to indicate different INP sources for the results of each INP parameterization (scheme) in Fig. 13. However, this is not ideal if we are also using the color to scale the ice nucleation temperature. Thus, we provide supplemental figures presenting the results of different INP sources for each INP parameterization.

Now, a new sentence as below is added in the revised manuscript to address the influence of INSEKT sampling on the INP parameterization results: "*The overall consistent trend between INSEKT and PINE data clusters in Fig. 13 and*

*Fig. 14 for all INP parameterizations suggests that filtered INSEKT dataset does not show discrepancy compared to PINE dataset and not influence the INP parameterization development.*"

4. **Section 3.4.2 and Table 3: An explanation about the selected specific forms for each equation is needed. Also please describe how they were fitted to the measurements. This could be provided in the supplemental material. Also please revise the formulations and check for any potential typo. Given the number of parameters, it would be very unfortunate that potential typos are propagated into future modeling studies.**

We agree. We now add new statements in Section 3.4.2 to explain the differences between our new INP parameterizations and those provided in the literature, as detailed below:

"*We also adapt Tobo2013FBAP with new parameters and it will be compared to a new parameterization we develop, termed "Helmos Fluo$_{WIBS}$" that predicts INP as a function of Fluo$_{WIBS}$, WIBS$_{ratio}$ (Fluo$_{WIBS}$ to NonFluo$_{WIBS}$) and T. Compared to Tobo2013FBAP, the new factor "(WIBS$_{ratio}$)$^{(eT+f)}$" in "Helmos Fluo$_{WIBS}$" is used to capture the contribution of fluorescent particles to the observed INPs at different temperatures.*"

"*Given that Fluo$_{WIBS}$ may not include all potential INPs (especially for T < −20 °C where nonbiological particles dominate), we propose two parameterizations ("Helmos Total$_{WIBS\_1}$" and "Helmos Total$_{WIBS\_2}$") using Total$_{WIBS}$ to represent aerosol particles that may serve sources for INPs. Both parameterizations depend on Total$_{WIBS}$, WIBS$_{ratio}$ and T but with different formula forms. Compared to DeMott2015 and DeMott2010, "Helmos Total$_{WIBS\_1}$" and "Helmos Total$_{WIBS\_2}$" also consider the effect of fluorescent particle portioning in different INP sources by using the corresponding factor including "WIBS$_{ratio}$".*"

"*... two parameterizations ("Helmos Total$_{SMPS+APS}\_1$" and "Helmos Total$_{SMPS+APS}\_2$" using a similar formula to that of DeMott2015 and DeMott2010 respectively) are proposed to calculate N$_{INP}$ based on Total$_{SMPS+APS}$ and T.*"

Also, a paragraph as below is now added in the supplementary Section S9 to describe how the parameters in each INP parameterizations are calculated by fitting the observed dataset to the formula.

"*Nonlinear regression with robust fitting option was used to calculate parameters for each proposed parameterization. Bisquare robust fitting was used for the regression model function to minimize a weighted sum of squares (to minimize the effect of outliers – and advantage over the least-squares approach). Weighted least square options were also suggested in the recent literature for INP parameterizations (Li et al., 2022). Also, fitted parameters were calculated with the error term normally distributed with mean 0 and standard deviation ≤0.1.*"

We thoroughly checked the parameters in Table 3 - thank you for the suggestion!

5. **Line 463: change to "This means anthropogenic pollution may impact (HAC)[2] even when it is in the FT"**

Indeed so! The sentence now reads: "*This means anthropogenic pollution may impact (HAC)[2] even when it is in the FT.*"

6. **Figure 9. Typo in the x axis of 9a (23.9 instead of 2.39)**

Good point! Reviewer #1 also raised this point (minor comment #7), and changes were made.

7. **Line 785: revise this sentence. It is not clear.**

Good point. The sentence now reads as: "*In addition, we note that INSEKT INPs evaluated as lower estimations beyond a factor of 5 by comparing to PINE INPs (see Supplement S3) are excluded from the parameterization dataset.*"

8. **Conclusions: how the developed parameterizations could be applied in current climate models? Some models already include dependencies on dust, marine organics and PBAPs. A detailed**

**perspective/recommendation in this sense would be extremely useful for modelers and make even more applicable the results of this paper.**

The relevant statements are revised, as follows:

"*Existing INP parameterizations in the literature may be improved by including the ratio of fluorescent-to-nonfluorescent or coarse-to-fine particles if available. Firstly, regional models, like mesoscale Weather Research and Forecasting (WRF) model, can use the improved INP parameterizations to calculate the ice formation processes for MPCs which ultimately helps to predict the local weather condition changes, such as storm formation and evolution (Georgakaki et al., 2024). Moreover, published dataset from existing field studies can be reorganized to a large and inclusive data base for the development of more general INP parameterizations for MPCs in global scales. In doing so, the predictability and applicability of the developed INP parameterizations for climate models can be improved. In particular, MPC simulations for regions with different INP sources, such as a cross-road of different air masses like (HAC)$^2$, or for different regions with different prevailing INP sources, such as continental or marine regions, can be better achieved. Ultimately, the regional and global simulations of aerosol-cloud interactions, as well as climate modelling, can be benefited from the improved INP parameterization by using its own dataset.*"

**References:**

Conen, F., Rodríguez, S., Glin, C. H., Henne, S., Herrmann, E., Bukowiecki, N., and Alewell, C.: Atmospheric ice nuclei at the high-altitude observatory Jungfraujoch, Switzerland, Tellus B Chem. Phys. Meteorol., 67, https://doi.org/10.3402/tellusb.v67.25014, 2015.

Li, G., Wieder, J., Pasquier, J. T., Henneberger, J., and Kanji, Z. A.: Predicting atmospheric background number concentration of ice nucleating particles in the Arctic, Atmospheric Chemistry and Physics Discussions, https://doi.org/10.5194/acp-2022-21, 2022.

McCluskey, C. S., Ovadnevaite, J., Rinaldi, M., Atkinson, J., Belosi, F., Ceburnis, D., Marullo, S., Hill, T. C. J., Lohmann, U., Kanji, Z. A., O'Dowd, C., Kreidenweis, S. M., and DeMott, P. J.: Marine and Terrestrial Organic Ice-Nucleating Particles in Pristine Marine to Continentally Influenced Northeast Atlantic Air Masses, J. Geophys. Res. Atmos., 123, 6196-6212, https://doi.org/10.1029/2017jd028033, 2018.

Niemand, M., Möhler, O., Vogel, B., Vogel, H., Hoose, C., Connolly, P., Klein, H., Bingemer, H., DeMott, P., Skrotzki, J., and Leisner, T.: A Particle-Surface-Area-Based Parameterization of Immersion Freezing on Desert Dust Particles, J. Atmos. Sci., 69, 3077-3092, https://doi.org/10.1175/jas-d-11-0249.1, 2012.

Schneider, J., Höhler, K., Heikkilä, P., Keskinen, J., Bertozzi, B., Bogert, P., Schorr, T., Umo, N. S., Vogel, F., Brasseur, Z., Wu, Y., Hakala, S., Duplissy, J., Moisseev, D., Kulmala, M., Adams, M. P., Murray, B. J., Korhonen, K., Hao, L., Thomson, E. S., Castarède, D., Leisner, T., Petäjä, T., and Möhler, O.: The seasonal cycle of ice-nucleating particles linked to the abundance of biogenic aerosol in boreal forests, Atmos. Chem. Phys., 21, 3899-3918, https://doi.org/10.5194/acp-21-3899-2021, 2021.

Tobo, Y., Prenni, A. J., DeMott, P. J., Huffman, J. A., McCluskey, C. S., Tian, G., Pöhlker, C., Pöschl, U., and Kreidenweis, S. M.: Biological aerosol particles as a key determinant of ice nuclei populations in a forest ecosystem, J. Geophys. Res. Atmos., 118, 10,100-110,110, https://doi.org/10.1002/jgrd.50801, 2013.

Ullrich, R., Hoose, C., Möhler, O., Niemand, M., Wagner, R., Höhler, K., Hiranuma, N., Saathoff, H., and Leisner, T.: A New Ice Nucleation Active Site Parameterization for Desert Dust and Soot, J. Atmos. Sci., 74, 699-717, https://doi.org/10.1175/jas-d-16-0074.1, 2017.

Wieder, J., Mignani, C., Schär, M., Roth, L., Sprenger, M., Henneberger, J., Lohmann, U., Brunner, C., and Kanji, Z. A.: Unveiling atmospheric transport and mixing mechanisms of ice-nucleating particles over the Alps, Atmos. Chem. Phys., 22, 3111-3130, https://doi.org/10.5194/acp-22-3111-2022, 2022.

---

## Author Comment (AC3)

**Response to EGUSPHERE-2024-511 reviews for RC3**

We thank Reviewer #3 for their effort and feedback on our manuscript EGUSPHERE-2024-511. In response to the suggestions and questions, please find our answers and corrections listed below: **Reviewer #3 comments are extracted in bold from original review supplement**; our responses are given directly below in normal font; *the original text in previous manuscript is repeated in red italic* and *revised text is typed in blue italic*.

**General comments:**

**I read the manuscript and found it extremely interesting, well organized, clearly written and scientifically sound. The manuscript presents and analyses an extremely valuable observational dataset of INP properties and ancillary information, useful for interpreting INP sources under different circulation regimes in the Mediterranean climate hotspot. I recommend publication, once the following minor comments are addressed.**

**If I have to make a remark, the authors compare their results with a limited number of previous studies reporting INPs in the Mediterranean basin. Considering that the literature on the subject is not huge, they could have done something more thorough. Nevertheless, I understand that the paper is already long as it is and that the authors selected for comparison the available immersion freezing INP datasets for major comparability with the INSEKT data.**

**A second remark would regard the paper length. Maybe the authors could have chosen a different publication strategy to disseminate their data (two shorter publications, instead of this very large one).**

We thank Reviewer #3 for their enthusiastically positive comments. We considered different publication strategies but concluded it would be best to have everything in one place. In terms of comparison, we decided to focus on the Mediterranean region – a future study could certainly focus on a more global comparison.

**Minor comments**

1. **L280-281. "the distinction of marine aerosols above PBL is indicated by the highest Cl- fraction compared to the other scenarios": the motivation for this is not clear. Please, explain or support with an appropriate citation.**

Good point! The revised sentence now reads as: "*In addition, the distinction of marine aerosols above PBL is indicated by the highest Cl⁻ fraction* range compared to the other scenarios *not under the FT condition (Fig. 3f), given that high Cl⁻ concentration is reported as a character of marine aerosols (Xiao et al., 2018).*"

2. **Fig. 5. It is not clear what the x-axis represents. I guess time? Please add the axis scale and clarify better. Moreover, double check this sentence in the caption: "The fraction of each type of particle from an identified aerosol source is shown by the x-axis". Maybe it should be y-axis?**

We apologize for this lack of clarity. The issue was raised by the other reviewers too and now is addressed.

3. **L556. Remove the symbol #.**

Done

4. **L575. I have the feeling that the term "secondary" can be subject to misinterpretations given the different meanings that it can assume in aerosol literature (secondary aerosol, secondary ice formation…); maybe you could use the term "minor" to indicate a lower role of small-sized aerosol particles as INPs. This applies also in L655.**

Good point. The revised (relevant) sentences now read as:

"*In comparison to Total$_{APS}$, Coarse$_{APS}$, Fluo$_{WIBS}$ (Fig. 9) or Total$_{WIBS}$ (Fig. S7), the smaller R and ρ values for the relation between Total$_{SMPS+APS}$ and INPs also indicates that small-sized aerosol particles in the SMPS size range may play a minor role for serving as INPs compared to larger-sized particles measured in the APS and WIBS size range.*"

"*Such a difference suggests the INPs that are of a biological origin become less important when the overall IN ability and INP abundance of the source is higher, such as dust plumes, indicating a less pronounced role of biological particles in dust-containing sources in serving as INPs.*"

**5. L612. I am not sure that "wrt." Can be immediately understood by non-native speakers.**

Good point. It is now spelled out (as "*with respect to*") in the manuscript.

**References:**

Xiao, H.-W., Xiao, H.-Y., Shen, C.-Y., Zhang, Z.-Y., and Long, A.-M.: Chemical Composition and Sources of Marine Aerosol over the Western North Pacific Ocean in Winter, Atmos., 9, https://doi.org/10.3390/atmos9080298, 2018.